# FEOTS v0.0.0: A New Offline Code for the Fast Equilibration of Tracers in the Ocean

Joseph Schoonover[1], Wilbert Weijer[2], and Jiaxu Zhang[3,4]

[1]Fluid Numerics LLC., Boulder, CO 80304
[2]Los Alamos National Laboratory, Los Alamos, NM 87545
[3]Cooperative Institute for Climate, Ocean, and Ecosystem Studies, University of Washington, Seattle, WA 98105
[4]NOAA/Pacific Marine Environmental Laboratory, Seattle, WA 98115

**Correspondence:** Joseph Schoonover (joe@fluidnumerics.com)

**Abstract.** In this paper we introduce a new software framework for the offline calculation of tracer transport in the ocean. The Fast Equilibration of Ocean Tracers Software (FEOTS) is an end-to-end set of tools to efficiently calculate tracer distributions on a global or regional sub-domain using transport operators diagnosed from a comprehensive ocean model. To the best of our knowledge, this is the first application of a Transport Matrix Model to an eddying ocean state. While a Newton-Krylov-based equilibration capability is still under development and not presented here, we demonstrate in this paper the transient modeling capabilities of FEOTS in an application focused on the Argentine Basin, where intense eddy activity and the Zapiola Anticyclone lead to strong mixing of water masses. The demonstration illustrates progress in developing offline passive tracer simulation capabilities, while highlighting the challenges of the Impulse Response Functions approach in capturing tracer transports by a non-linear advection scheme. Our future work will focus on improving the computational efficiency of the code to reduce time-to-solution, using different basis functions to better represent non-linear advection operators, applying FEOTS to a parent model with unstructured grids (MPAS-Ocean), and to fully implement a Newton-Krylov steady state solver.

## 1 Introduction

Many oceanographic research problems involve the transport and distribution of tracers that do not feed back on the ocean dynamics. Examples of such problems are the diagnostic tracking of water masses using passive tracers (e.g., Dukhovskoy et al., 2016; Zhang et al., 2021); validating the use of isotopes or grain size distributions in marine sedimentary records to infer past ocean circulation changes (e.g., Jahn et al., 2015; Zhang et al., 2017; Gu et al., 2019; Missiaen et al., 2020); assessing anthropogenic carbon uptake by the ocean (e.g., Sarmiento et al., 1992; Khatiwala et al., 2009; Wang et al., 2012); studying the evolution of marine biogeochemical systems (e.g., Séférian et al., 2020); or tracking the fate of microplastics in the ocean (e.g., Mountford and Morales Maqueda, 2019). In many cases, the transport of these tracers is dominated by mesoscale processes like eddies. Typical ocean climate models use grids that are too coarse to explicitly resolve these processes and rely on parameterizations to simulate their impact on tracer fields; but it has become clear that these eddy-parameterized models fail to reproduce some critical aspects of the real ocean (e.g., Lozier, 2010). What is more, the ocean also contains *dynamical* features that rely on mesoscale eddies, and which cannot be reproduced by low-resolution models.

A case in point is the Argentine Basin. It is among the most turbulent regions in the World Ocean (Fu and Smith, 1996), mostly on account of the confluence of two western boundary currents, the Brazil and Malvinas Currents (e.g., Garzoli, 1993). A seamount in the center of the basin is associated with a local minimum in eddy kinetic energy (Fu and Smith, 1996), but is surrounded by a very strong barotropic vortex, the Zapiola Anticyclone (ZA; Saunders and King, 1995; de Miranda et al., 1999). This anticyclone is understood to be driven by the intense eddy field (Dewar, 1998), and has been shown to inhibit exchanges between the interior of the ZA and its surroundings (Weijer et al., 2015, 2020). Strikingly, the Argentine Basin is also a main conduit of water mass exchange between the Atlantic and Southern Oceans (e.g., Jullion et al., 2010), so eddy-driven mixing of water masses in the Argentine Basin may have implications for the global thermohaline circulation.

It is clear that studying tracer transport and mixing in the Argentine Basin requires an ocean model that resolves the ocean's mesoscale; not just to accurately represent the transports by narrow boundary currents and the turbulent eddy field, but also to generate the ZA in the first place. Second, the equilibration of tracers at deeper levels may take many decades or centuries, requiring a model capability that can be run for such long times, or can determine equilibria directly using iterative solvers. Third, this problem is ideally addressed using a representation of the global ocean circulation; not only so that the ocean circulation in the region of interest is fully consistent with large-scale oceanographic drivers, but also for the practical reason that not every regional problem would require a unique model configuration.

The cost and technical challenges of running global and fully-dynamic ocean models make it expensive and difficult at best, but often impossible, to address problems like these. It is obvious that there is a need for simple and efficient tools that solve tracer transport and distribution problems, without the need to explicitly simulate ocean dynamics. In the past decade or so, several Transport Matrix Models (TMMs) have been developed that allow for the simulation of passive tracers in a stand-alone (offline) code, using transport operators that have been diagnosed from comprehensive ocean models (Primeau, 2005; Khatiwala et al., 2005; Khatiwala, 2007; Bardin et al., 2014; Kvale et al., 2017; Zanna et al., 2019; Chamberlain et al., 2019). In particular, Khatiwala et al. (2005) pioneered the approach of empirically estimating an ocean model's transport processes by diagnosing the action of the transport (advection and diffusion) operators on simple basis functions like impulse fields. The resulting Impulse Response Functions (IRFs) can straightforwardly be converted to the desired transport matrices. This process requires minimal intervention in the parent model.

Despite the success of these techniques, TMM models have generally been applied to low-resolution, non-eddying ocean states only. This makes them unsuitable to study problems where eddies play a zeroth-order role in tracer transport and ocean dynamics, like the Argentine Basin. To the best of our knowledge, the capability introduced in this paper is the first TMM model developed for and applied to eddying ocean states, with transport operators diagnosed from a global ocean model with nominal resolution of $0.3°$ and 100 vertical levels ($\sim 10^8$ degrees of freedom).

Our computational framework is embodied in the Fast Equilibration of Ocean Tracers Software (FEOTS; https://github.com/LANL/FEOTS). FEOTS is based on the methodology of Bardin et al. (2014) but is specifically designed to tackle the large computational problems associated with tracer transport in a global eddying ocean. In particular, FEOTS is written in Object-Oriented Fortran instead of Matlab, which makes it straightforward to port the code to supercomputers, to parallelize the workflow, and to expose the code to a plethora of existing solver libraries. Also, FEOTS provides an end-to-end set of tools

that streamlines the process of building and running an offline tracer model. It i) uses an advanced optimization algorithm to generate an optimal set of impulse functions, given the grid layout and operator stencils for the parent model; ii) transforms the resulting IRFs obtained from the parent model to transport operators; iii) sets up regional or global tracer problems, with different types of tracers; iv) runs forward simulations of the offline tracer model; and v) uses a Newton-Krylov solver to determine steady tracer distributions. To date, the authors have implemented the first four of these features, and their implementation, validation, and verification are presented in this paper. We plan to fully implement a Newton-Krylov steady state solver in future releases. The framework is applied here to the Parallel Ocean Program (POP; Smith et al., 2010); but the design is flexible enough to be generally applicable –in particular to the new generation of ocean models with unstructured grids, like the Ocean Model for Prediction Across Scales (MPAS-Ocean, Ringler et al., 2013). Other future work aims to improve time-to-solution by exposing data parallelism in forward simulation components of FEOTS.

In this paper we present validation and verification results for the FEOTS offline tracer solver in a regional forward simulation configuration focused on the Argentine Basin. Specifically, we will show that uniform tracer fields are preserved within $0.01\%$ after five years of offline model integration and we will present a comparison of the offline regional tracer simulation using five-day averaged transport operators with an online tracer simulation using the parent model. Finally, we comment on the compute costs for running offline simulations with FEOTS

## 2 Methodology

Our work builds on the methodologies of Bardin et al. (2014) to create FEOTS. FEOTS comes with tools to generate impulse fields, translate impulse response fields to sparse matrices corresponding to advection and lateral diffusion, create vertical diffusion operators from eddy diffusivities reported by the parent model, and execute offline regional transient tracer simulations. In section 2.1 we discuss the parent model POP, which is used for this study. In section 2.2 we present the governing equations for a non-interacting passive-tracer system and outline the methodology for capturing transport operators from a comprehensive ocean model in 2.3. The challenge of using a flux-limited advection scheme is discussed in section 2.4. The offline forward stepping algorithm and treatment of vertical mixing is presented in section 2.5. Last, the constant preservation test problem is defined in section 2.6 and the Argentine Basin test problem is defined in 2.7.

### 2.1 Parent Model

FEOTS is used for offline tracer simulations using transport operators diagnosed from a comprehensive ocean model, the "parent model". Here we apply FEOTS to the Parallel Ocean Program, in the framework of the E3SMv0-HiLAT climate model (Hecht et al., 2019). Our specific configuration is described in Zhang et al. (2019) and is referred to as E3SMv0-HiLAT03. E3SMv0-HiLAT03 has a tripole grid with nominal 0.3° spatial resolution. The grid has 1200x800 grid cells, and 100 levels in the vertical. The grid has a 'seam' in the Arctic that connects the poles in Siberia and in Canada. Although technically not eddy-resolving in most of the World Ocean (Hallberg, 2013), this configuration has a vigorous eddy field (Zhang et al., 2019), and a realistic representation of the eddy-driven Zapiola Anticyclone in the Argentine Basin (Weijer et al., 2020).

With $\mathbf{O}(10^8)$ degrees of freedom, and an eddying field that requires transport operators at high temporal frequency (e.g., 5-daily averages), the data volume of the diagnosed IRFs can become unmanageable quickly. It is therefore critical to keep the number of IRFs to an absolute minimum. This means that we need to choose advection and diffusion treatments that have the most compact stencils. Typically high-resolution ocean models use a bi-harmonic mixing scheme to dampen out the dispersive errors caused by the advection operator (e.g., Hecht et al., 2008). Bi-harmonic mixing is clearly undesirable for our application, given its huge stencil. Of the three advection schemes currently implemented in POP (Smith et al., 2010), the centered and $3^{rd}$-order upwind schemes require explicit diffusion to manage the dispersion error. The flux-limited Lax-Wendroff scheme does not, making this the only reasonable choice for our application; even though its stencil is quite large (27 grid points, $3 \times 3 \times 3$) compared with the other two (7 grid points for centered, 13 for the $3^{rd}$-order upwind scheme). This choice requires 53 impulse functions to capture the advection operator (compared to 34 impulse functions for the $3^{rd}$-order upwind scheme), but eliminates the need for explicit diffusion.

The model is forced by the normal-year CORE-II climatology (Coordinated Ocean-Ice Reference Experiments version 2; Griffies et al., 2012), which has been a widely used framework to force ocean and/or sea ice models for hindcast simulations. With a time step of 7 minutes, the model typically yields maximum CFL values of $\mathbf{O}(10^{-1})$ or smaller. Although the model was run for 186 years, we diagnosed the transport operators for the 5-year period starting at simulation year 64. Even though 63 years of spin-up is not sufficient to fully equilibrate the stratification in the deep ocean, the main circulation features (e.g., boundary currents, the eddy field, the Zapiola Anticyclone) are well established by then, making this an appropriate data set to demonstrate the capability of FEOTS. We refer to Weijer et al. (2020) for evaluation of the hydrography and circulation in the Argentine Basin in a companion simulation.

## 2.2 Governing Equations

We model a passive dye tracer as a concentration field that is subjected to advection and diffusion,

$$\frac{\partial}{\partial t}[(1+v)c] + \nabla \cdot (\boldsymbol{u}c - \mathbb{K}\nabla c) = 0 \tag{1}$$

where $v$ is the fluid volume anomaly, $c$ is the tracer concentration, $t$ is time, $\boldsymbol{u}$ is the ocean velocity field, and $\mathbb{K}$ is the diffusivity tensor that models unresolved eddy activity. The fluid volume anomaly is a unitless quantity that is a measure of the relative change of the fluid volume due to movement of the free-surface. This formulation is chosen so that total tracer is conserved and so that the offline model is consistent with the parent model (Smith et al., 2010).

In practice, the fluid volume anomaly is defined as the ratio of the free surface height to the upper most grid cell, only in the upper most grid cell,

$$v = \frac{\eta}{dz_1}\delta_{k,1} \tag{2}$$

where $\eta$ is the free surface height, $dz_1$ is the grid cell thickness in the upper most grid cell, and $\delta_{k,1}$ is the Kronecker delta function.

The initial and boundary conditions are set to be

$$c(t = 0) = c_0(z, \theta, \phi) \tag{3}$$

and

$$c = c_b(z, \theta, \phi, t), \tag{4}$$

where $z$ is depth (measured positive downward), $\theta$ is longitude, and $\phi$ is latitude. The semi-discrete form of (1) can be written as:

$$\frac{\partial}{\partial t}[(1+v)c] + (A + D_h + D_v)\, \boldsymbol{c} = 0 \tag{5}$$

where $\boldsymbol{c}$ is a vector of the discrete values of the tracer, $A$ is the advection matrix, $D_h$ is the horizontal diffusion matrix, and $D_v$ is the vertical diffusion matrix. In the examples presented in this paper, the advection matrix corresponds to the Flux-Limited Lax-Wendroff advection scheme on an Arakawa B-Grid; this scheme is chosen in the parent model configuration. Since we do not enable explicit lateral tracer diffusion in the parent model in this study, all elements of $D_h$ are zero. To create the vertical diffusion matrix, we diagnose diffusivity coefficients, which are generated from the KPP parameterization, and create a vertical mixing operator offline using centered differencing with no-flux conditions applied at the ocean free surface and at the sea-floor.

## 2.3   Graph Coloring approach to Operator Diagnosis

The purpose of capturing the impulse response functions is to diagnose sparse matrices that are consistent with the advection discretization in the parent model. To illustrate this procedure, suppose that the parent model has $N$ grid cells and that $N$ impulse fields are set as the Kronecker delta functions:

$$[\boldsymbol{c}^{(i)}]_{i=1}^{N} = [\delta_{j,i}]_{i=1}^{N} \tag{6}$$

In this setup, impulse function $i$ is zero at all grid cells except for grid cell $i$ where the impulse function has a value of one. Application of the transport matrix to each of the $\boldsymbol{c}^{(i)}$ returns column $i$ of $A$,

$$A\boldsymbol{c}^{(i)} = \sum_{j=1}^{N} A_{m,j}\delta_{j,i} = A_{m,i} \quad \text{for } m = 1, 2, 3, ..., N \tag{7}$$

While using a set of Kronecker delta functions will completely diagnose all of the elements of the transport matrix, this strategy is computationally expensive. For each time step, this strategy requires computing the advective tendency for $N$ tracer fields, where $N$ is the number of grid cells. For example, coarse resolution model at $\mathbb{O}(1°)$ resolution have roughly $10^6$ grid cells. Storing $10^6$ impulse and impulse response functions would require approximately 3 TB of memory at double precision.

To reduce the number of required impulse functions to fully diagnose the transport matrices, we can take advantage of the fact that the advection scheme results in a sparse matrix. Equivalently, the domain of influence of the advection operator is

limited to nearby grid cells. The parent model employed in Bardin et al. (2014) used a third order upwind scheme, where the impulse response is guaranteed to extend no further than two grid-cells in each spatial dimension, giving a 5x5x5 brick for the domain of influence. Because of this, the authors used a set of 125 tracer fields,

$$c(i,j,k;i_0,j_0,k_0) = \delta_{i0,i(mod5)}\delta_{j0,j(mod5)}\delta_{k0,k(mod5)} \ \text{ for } i_0 = 1,...,5; j_0 = 1,...,5; k_0 = 1,...,5 \tag{8}$$

FEOTS offers a unique capability to generate a minimal set of impulse functions by posing the problem as a graph coloring problem. A graph $G(V,E)$ is defined by a set of vertices $V$ and edges $E$ that connect the vertices. Two vertices connected by an edge are said to be adjacent. A valid graph coloring of $G(V,E)$ assigns colors to each vertex so that no two adjacent vertices have the same color. To calculate impulse functions that can be used to diagnose transport operators, FEOTS offers functionality to express a POP mesh and an advection stencil into an equivalent graph that is colored with a Greedy algorithm. This formulation has the benefit that it can be generalized to parent models based on unstructured grids and it takes into account irregular boundaries from variable bathymetry.

In FEOTS, graph vertices $V$ correspond to each ocean grid cell, centered on tracer points, in the POP mesh. Two vertices are adjacent if their impulse response functions overlap. Because a valid coloring results in adjacent vertices having distinct colors, vertices with the same color can safely be assigned to the same impulse function. Consequently, the chromatic number of the graph corresponds to the number of impulse functions used for model diagnosis. For this work, the parent model uses a $0.3°$ periodic tripole mesh and the 3rd order flux-limited Lax-Wendroff advection scheme. This approach results in 53 impulse functions required to uniquely diagnose the transport operators.

The transport operators in equation (5) are diagnosed empirically from the parent model, using the methodology used by Bardin et al. (2014) and pioneered by Khatiwala et al. (2005). This process involves diagnosing and time averaging the impulse response functions corresponding to each of the 53 impulse fields in the passive tracer equations in POP. Modifications are made in POP to initialize the passive tracers to the impulse functions at the beginning of each time step. The impulse response function is set equal to the advective tendency as diagnosed in POP. After diagnosing the IRF in each time step, the tracer field is reset to the impulse field so that the IRF at the next time step is saved. The IRFs are time-averaged over a configurable averaging period and written out to file.

For each time averaged IRF, we create a sparse matrix representation of the advection operator in POP. Modeling the advection operator as a matrix-vector multiplication, equation (7) shows that passing an impulse at grid point $i$ through the advection operator, returns column $i$ of the matrix that corresponds to the advection. We use this to construct a sparse matrix in compressed-row-storage format that corresponds to the advection operator.

For our test problems, we diagnosed the 5-day averaged IRFs and vertical diffusivities for the 5-year analysis period of the parent model. We repeated the simulation for 105 days, diagnosing 1-day averaged IRFs. With this methodology and the 7-minute time step, the one-day averaged operators are each an average of 1440 IRF snapshots and the five-day averaged operators are each an average of 7200 IRF snapshots. The data volume of the global parent model five years' worth of 5-day averaged operators (365 IRFs and diffusivities) is about 9 TB. Once transformed to transport operators, the data volume is 4 TB.

## 2.4 Flux-limited advection

The parent model uses a nonlinear Flux-Limited Lax-Wendroff advection scheme; the flux limiter is equivalent to the ULTIMATE flux-limiter described in Leonard (1991) and Hundsdorfer and Trompert (1994). Because FEOTS treats offline advection as a linear operation, online advection with the parent model, in this case, can not be equivalent. In what follows, we will illustrate how the impulse functions result in the diagnosis of a more diffusive advection scheme than Lax-Wendroff.

In POP, the tracer equations are discretized using a finite volume approximation on an Arakawa B-grid. The non-linear Flux-Limited Lax-Wendroff scheme is applied in three-dimensions using a dimensional splitting technique, the details of which can be found in Chapter 6 of Smith et al. (2010). For the purpose of this discussion, it is sufficient to consider the one-dimensional problem of calculating the advective flux at a tracer cell face. The advective flux at a tracer cell face can be written as

$$F_f = u_f \left[ c_u + \Psi(r)(c_d - c_u) \right] \tag{9}$$

where $F_f$ is the advective flux at a tracer cell face, $u_f$ is the normal velocity at a tracer cell face, $c_u$ corresponds to the tracer concentration in the upwind direction, $c_d$ corresponds to the tracer concentration in the downwind direction, and $\Psi(r)$ is the limiter function where $r$ is a measure of the monotonicity of $c$. The monotonicity in this method is defined as

$$r = \frac{c_u - c_{u-1}}{c_d - c_u} \tag{10}$$

where $c_{u-1}$ is the tracer concentration one cell further in the upstream direction. When $r < 0$, the tracer concentration has a local maximum or minimum centered around $c_u$. For the ULTIMATE flux limiter, $\Psi(r) = 0$ when $r < 0$, in which case the flux evaluates to the upwind flux.

The impulse functions used in this study all have local maximum centered at the impulse locations. At the impulse locations, $r < 0$, and therefore the diagnosed flux is the more diffusive upwind flux. This behavior will be apparent and quantified when comparing the online and offline simulations in subsection 3.2.

## 2.5 Time integration

Forward integration of the offline tracer model uses Backward Euler for vertical mixing and can use Forward Euler, Adams-Bashforth $2^{nd}$ Order, or Adams-Bashforth $3^{rd}$ Order for transport. As in Bardin et al. (2014), we forward step an equation for the volume anomaly using a forward Euler method. Volume anomalies arise due to divergence in the transport field at the upper-most z-level that are associated with fluctuations of the free-surface.

In general, the time integration scheme can be written as

$$\boldsymbol{v}^{n+1} = \boldsymbol{v}^n + \Delta t \mathbb{A}^n \boldsymbol{i} \tag{11}$$

$$(\mathbb{I} + \mathbb{V}^{n+1} + \mathbb{D}_v)\boldsymbol{c}^{n+1} = (\mathbb{I} + \mathbb{V}^n)\boldsymbol{c}^n + \Delta t(\mathbb{A} + \mathbb{D}_h)\boldsymbol{c}^* \tag{12}$$

where $\boldsymbol{v}$ is the volume anomaly, $\boldsymbol{i}$ is a constant vector whose elements are all set to one, $\boldsymbol{c}^*$ depends on the time integration scheme that is used (Table 1) and $\mathbb{V}^{n+1}$ is a diagonal matrix whose diagonal elements are the volume anomalies.

| Method | $c^*$ |
|---|---|
| Forward Euler | $c^n$ |
| Adams-Bashforth 2nd Order | $\frac{3c^n - c^{n-1}}{2}$ |
| Adams-Bashforth 3rd Order | $\frac{23c^n - 16c^{n-1} + 5c^{n-2}}{12}$ |

**Table 1.** Optional values for $c^*$ in Eq. (12), based on the choice in time integration scheme. 3rd Order Adams-Bashforth is used for the Argentine Basin test problem presented in this paper.

In ocean models, advection and horizontal diffusion operators have a compact stencil, enabling the use of the IRF approach
described in the previous section to 'capture' these advection and diffusion operators. However, vertical diffusion is usually treated differently, by solving a tri-diagonal system that touches the entire water column. The reason is that high values of vertical diffusivity are applied where the water column is unstable; and these values easily render any explicit scheme unstable. Consequently, as the region of influence of the vertical diffusion operator is the entire water column, the IRF approach would demand a separate IRF field for each vertical level of the model grid, which is prohibitive for finely resolved grids. Instead,
FEOTS treats the vertical solve similarly as the parent model, so rather than IRFs, the vertical diffusivities are diagnosed, saved, and used to recreate the vertical diffusion operators offline.

Forward stepping FEOTS requires inverting a tri-diagonal system of equations, given by equation (12), for the tracer concentration in order to incorporate vertical mixing. To solve this system, we use the preconditioned conjugate gradient algorithm (Shewchuk, 1994) with a diagonal preconditioner. The initial solution guess for the vertical mixing solver is set as the tracer
concentration that is predicted without vertical mixing,

$$c_0^{n+1} = (\mathbb{I} + \mathbb{V}^{n+1})^{-1}((\mathbb{I} + \mathbb{V}^n)c^n + \Delta t(\mathbb{A} + \mathbb{D}_h)c^*) \tag{13}$$

For the results presented in this paper, we use the 3rd Order Adams-Bashforth time integrator and the conjugate gradient solver is stopped when the residual magnitude, relative to the initial solution guess magnitude, is less than $10^{-6}$. We use a 15 minute time step, and a typical maximum CFL value, obtained by eigenvalue analysis of the transport operators, is $\mathbf{O}(0.1)$.

**2.6 Constant Preservation Test Problem**

We now show that constant preservation is expected in the discrete system. This provides the foundation for a test case that we use to verify the FEOTS implementation. From Equations (11) and (12), we start by assuming that there is no lateral or vertical diffusion and the initial tracer field is a constant value of 1 ($c^0 = i$). The discrete system at the initial time step is then,

$$v^1 = v^0 + \Delta t \mathbb{A} i \tag{14a}$$

$$(\mathbb{I} + \mathbb{V}^1)c^1 = (\mathbb{I} + \mathbb{V}^0)i + \Delta t \mathbb{A} i \tag{14b}$$

Note that we can write these equations using indicial notation, which makes it easier to obtain the result for $c^1$

$$v_j^1 = v_j^0 + \Delta t \sum_k \mathbb{A}_{j,k} \tag{15a}$$

$$c_j^1 + v_j^1 c_j^1 = 1 + v_j^0 + \Delta t \sum_k \mathbb{A}_{j,k} \tag{15b}$$

Substituting (15a) into (15b) gives

$$(1 + v_j^0 + \Delta t \sum_k \mathbb{A}_{j,k})c_j^1 = 1 + v_j^0 + \Delta t \sum_k \mathbb{A}_{j,k} \tag{16}$$

The only solution to (16) is $c_j^1 = 1$ for each $j$, implying that the discrete solution remains a constant. If we progress to the next time step, using Forward Euler, the same result is obtained. If instead, we switch to second order Adams Bashforth for the next time step, we have that

$$c_k^* = \frac{3c_k^1 - c_k^0}{2} = 1 \tag{17}$$

Using this for $c^*$ results in

$$(1 + v_j^2 + \Delta t \sum_k \mathbb{A}_{j,k})c_j^2 = 1 + v_j^1 + \Delta t \sum_k \mathbb{A}_{j,k} \tag{18}$$

which again implies that $c^2 = i$. This result also holds for 3rd order Adams-Bashforth in this case, since

$$c_k^* = \frac{23c^2 - 16c^1 + 5c^0}{12} = 1 \tag{19}$$

when $c^2 = c^1 = c^0 = i$.

When vertical mixing is introduced, the solution obtained in (16) serves as an initial guess to the conjugate gradient solver. If the initial guess for the solver is a constant tracer field,

$$\mathbb{D}_v c = 0 \tag{20}$$

This result in (20) is due to the use of a finite volume discretization for the vertical diffusion with no-flux conditions at the ocean surface and at the sea-floor. In exact arithmetic, we would therefore expect the tracer field to remain a constant.

This analysis shows that a constant tracer field remains a constant tracer field under the discretizations employed in FEOTS. Finite precision arithmetic, however, can produce slight deviations from a constant field, and the results presented in section 3.1 characterize the behavior of round-off errors for a constant tracer field, both with and without vertical mixing.

## 2.7 The Argentine Basin Test Problem

As discussed in the introduction, the Argentine Basin is an ideal region for testing a tracer transport capability in an eddying ocean model. In this paper we compare online and offline simulations of dye tracers that are initialized at the boundaries of

the Argentine Basin, here chosen as $[52.18^oS, 28.06^oS] \times [70.25^oW, 24.90^oW]$. In all simulations, the only source of tracers comes from the model boundary conditions applied along the southern, eastern, and northern boundaries.

We want to distinguish between water masses that originate from each of the domain boundaries and above and below 1000 m depth. This is accomplished by simulating six passive tracer fields $D_i$ with boundary conditions:

$$D_1 : c_b^1 = H(1000 - z) \quad \text{when } \phi = 52.18^oS \tag{21a}$$

$$D_2 : c_b^2 = H(z - 1000) \quad \text{when } \phi = 52.18^oS \tag{21b}$$

$$D_3 : c_b^3 = H(1000 - z) \quad \text{when } \theta = 24.90^oW \tag{21c}$$

$$D_4 : c_b^4 = H(z - 1000) \quad \text{when } \theta = 24.90^oW \tag{21d}$$

$$D_5 : c_b^5 = H(1000 - z) \quad \text{when } \phi = 28.06^oS \tag{21e}$$

$$D_6 : c_b^6 = H(z - 1000) \quad \text{when } \phi = 28.06^oS \tag{21f}$$

where $H(x)$ is the Heaviside step function. With this configuration, tracers $D_1$, $D_3$, and $D_5$ are released in the upper 1000 m, on the southern, eastern, and northern boundaries, respectively; while tracers $D_2$, $D_4$, and $D_6$ are released below 1000 m. Note that maximum mixed layer depths in the Argentine Basin in winter are around 500 m in this model, so deep convection should not play a role in the transport of these tracers across the 1000 m depth horizon.

## 3 Results

In this section, we present results of an offline regional study focused on the Argentine Basin. We first present verification of the constant preservation property (section 3.1). Then we present a comparison of the tracer simulation results in the parent model (online) and the offline model using 5-day averaged transport operators (section 3.2), followed by a comparison between 1-day and 5-day averaged transport operators (section 3.3). Finally, we will discuss the computational performance of the code 280 (section 3.4) on the systems where our simulations were conducted.

### 3.1 Constant Preservation

The parent model uses a finite volume discretization that guarantees the preservation of constant tracer fields. To verify that FEOTS accurately diagnoses transport operators that are representative of the parent model, our first simulation involves verifying that a constant tracer field remains constant under the action of the diagnosed transport operators.

In all of our simulations, we have opted to use single precision arithmetic and have enabled aggressive compiler optimizations (compiler option -*Ofast* with GCC 9.2.0). These choices were made to minimize data storage costs for the transport operators and post-processing output and to optimize the time-to-solution for the offline simulations. Although analytically we expect that the FEOTS algorithm should preserve constant tracer fields, errors from floating point arithmetic are expected to be the main source of constant preservation errors.

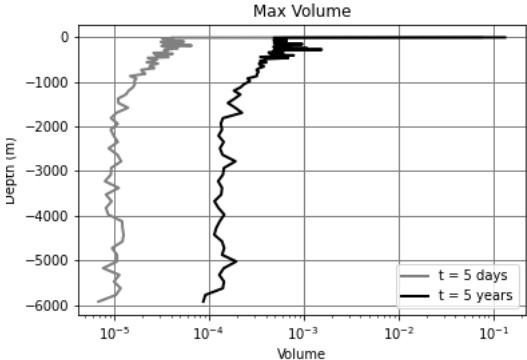

**Figure 1.** Vertical profile of the max volume anomaly after 5 days (gray) and after 5 years (black). In the exact form of the equations, the volume anomaly should only exist in the first vertical layer at the ocean surface. Non-zero values in the volume beneath the surface layer arise due to round-off errors.

To better understand the spatial distribution and the relative impact of round-off errors, we configure a simulation where the initial tracer field is set to $c_0 = 1$ and there is no external source or sink of tracer. Since $\nabla c_0 = 0$ and $\nabla \cdot \boldsymbol{u} = 0$ under the discretizations used in the parent model, we expect that $c_t = 0$ and $c = c_0 = 1$ for all time, as demonstrated in section 2.6.

Figure 1 shows the maximum volume anomaly in the domain as a function of depth after five days and five years of integration. Analytically, the volume anomaly is expected to be zero in all cells except the top-most layer. At the surface layer,

fluctuations in the free surface height are associated with non-zero fluid divergences that contribute to changes in the fluid volume. Beneath the surface layer, the fluid velocity field is expected to be divergence free. In general, larger errors in the volume anomaly are observed above 1000 m depth. After five days, errors in the deep ocean are $\mathbf{O}(10^{-5})$ and after five years, the deep ocean volume anomaly errors have grown by an order of magnitude to $\mathbf{O}(10^{-4})$. Larger errors are observed above 1000 m, reaching $\mathbf{O}(10^{-3})$ after 5 years. Note that the volume anomaly field is identical for all choices in time integrator for

the dye tracer and is independent of vertical mixing.

Errors in the volume anomaly lead to spurious values for predicted tracer concentrations. For this simulation, any deviation of the tracer concentration from its initial uniform value is erroneous. Figure 2 shows the max error in the dye tracer as a function of depth after five days and five years of integration, with and without mixing. After five years of integration, the maximum relative error with mixing is about $0.05\%$ and without mixing is about $0.01\%$. At depth, the errors in the tracer are

comparable with and without mixing. However, above 1000 m, particularly in the mixed layer, the inclusion of mixing results in an accumulation of round-off errors.

### 3.2 The Argentine Basin test case: Offline vs. Online Comparison

In addition to quantifying errors for the constant-tracer scenario, a practical concern is in the comparison between the online and offline tracer simulations. Do the tracer distributions simulated by FEOTS using transport operators diagnosed from the parent

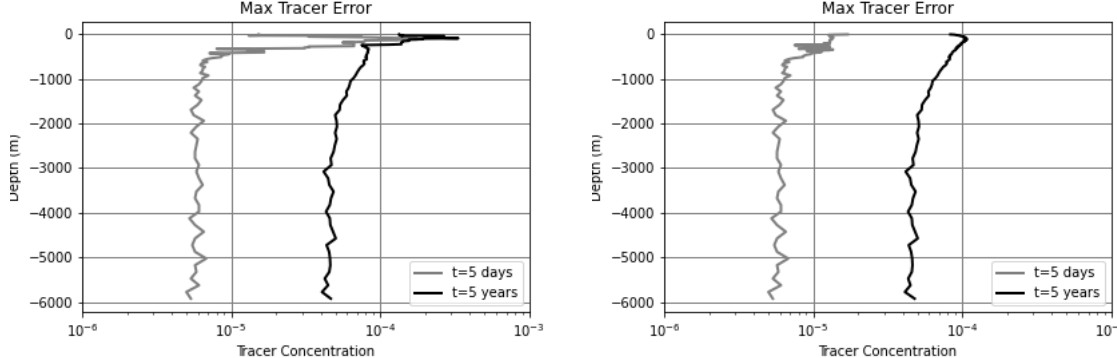

**Figure 2.** Vertical profile of the max error in the dye tracer after 5 days (gray) and after 5 years (black) for simulations with mixing (left) and without mixing (right). Errors that arise from integrating a uniform tracer field arise from round-off errors in volume anomaly. When mixing is enabled, vertical mixing redistributes errors and resulting in elevated errors in the upper layers in the simulation.

model E3SMv0-HiLAT03 (offline) faithfully represent the tracer distributions simulated by the parent model itself (online)? The time-averaging of the transport operators will introduce differences between the online and offline simulations, as will the flux-corrected advection operator. Our aim is to quantify and qualitatively describe the differences between a tracer simulation conducted directly in the online parent model and the offline model. We do this for our example problem, namely determining the source waters of the Zapiola Anticyclone.

Figure 3 compares the tracer fields obtained with the online and offline method, at the end of the 5-year analysis period. The dye tracers shown are those that are released in the upper 1000 m at each of the three domain boundaries. Visual inspection shows that the online simulation has sharper gradients compared to the offline simulation, even though the same dynamical features are visible. This is a result of the non-linear advection operator that is more diffusive when applied to impulse functions than to a smooth tracer field, as explained in section 2.4. The advective errors also affect integrated quantities of dye tracers

within the Zapiola Anticyclone. Figure 4 shows the vertical distribution of tracer concentrations averaged over the Zapiola Anticyclone, while Fig. 5 shows the total tracer stock.

Dye tracer $D_1$, sourced at the southern boundary in the upper 1000 m, reaches the ZA after about 45 days in the online simulation (Fig. 4, upper row). Concentrations in the upper 1000 m rise steadily in the first 2 years; then become mostly steady, indicating saturation is reached (Fig. 5a, black dashed). Just below the surface, concentrations in the final 3 years average

about 0.76, suggesting that 76% of the surface waters within the ZA may be derived from the Southern Ocean through the Malvinas Current. At 984 m this fraction is 0.45. Tracer stock below 1000 m increases slowly and contains 17% of the column stock after 5 years (Fig. 5a, black dotted). The offline simulation reproduces this behavior qualitatively, but with significant quantitative differences. $D_1$ reaches the ZA about 5 days earlier and increases faster than the online simulation, but upper-layer stock plateaus at a slightly lower level (Fig. 5a, gray dashed). Tracer fraction reaches 0.74 just below the surface, close to the

online simulation, but the saturation value of 0.37 at 1000 m depth is significantly lower. Figure 4 (upper row) shows that this

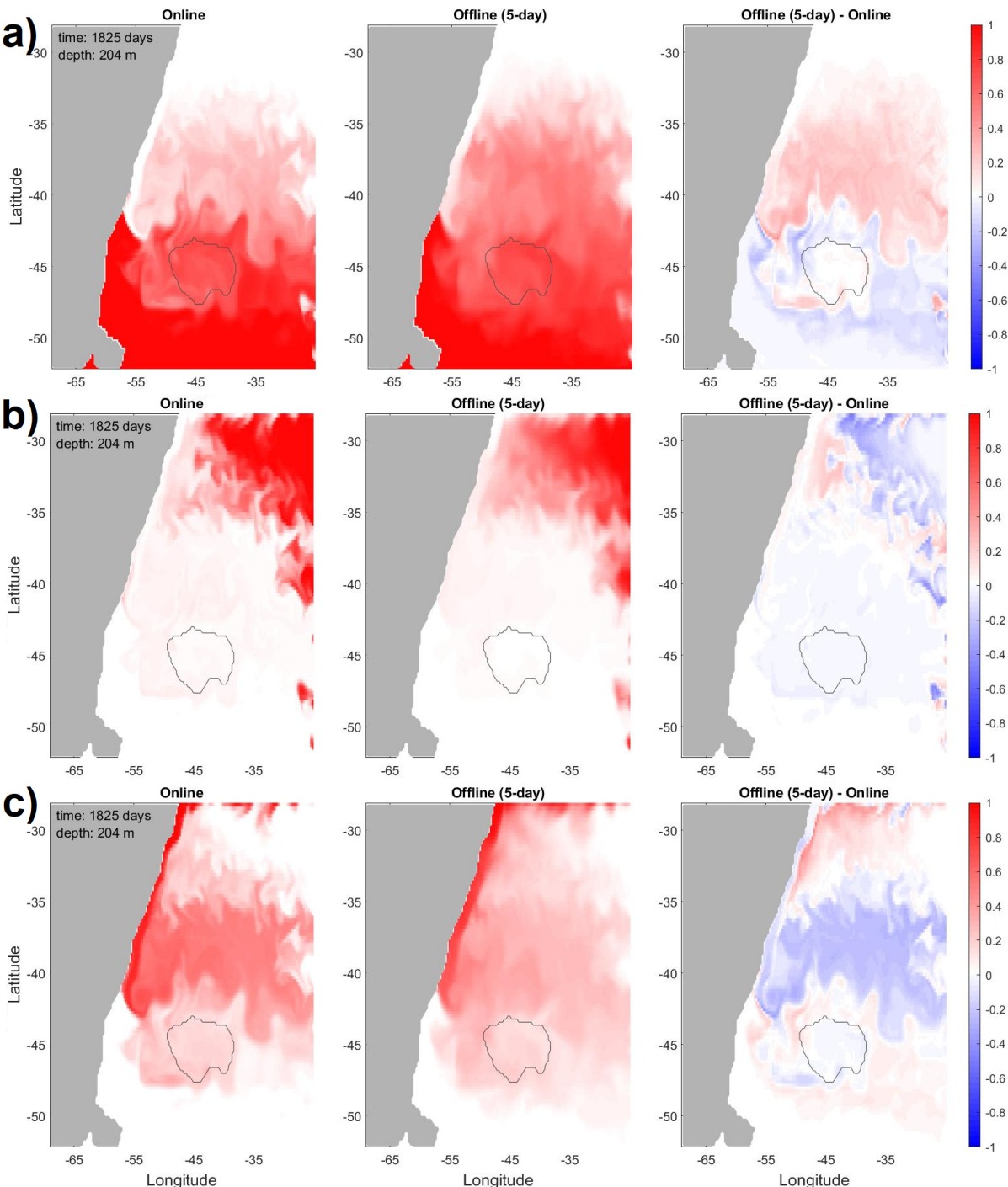

**Figure 3.** Online and offline tracer concentrations –and their difference– at 204 m depth, and at the end of the full 5-year period for which operators were diagnosed at 5-daily averages. Shown are dye tracers a) $D_1$ (sourced at the southern boundary), b) $D_3$ (eastern boundary), and c) $D_5$ (northern boundary); all sourced in the upper 1000 m. Gray contour indicates the location of the Zapiola Anticyclone.

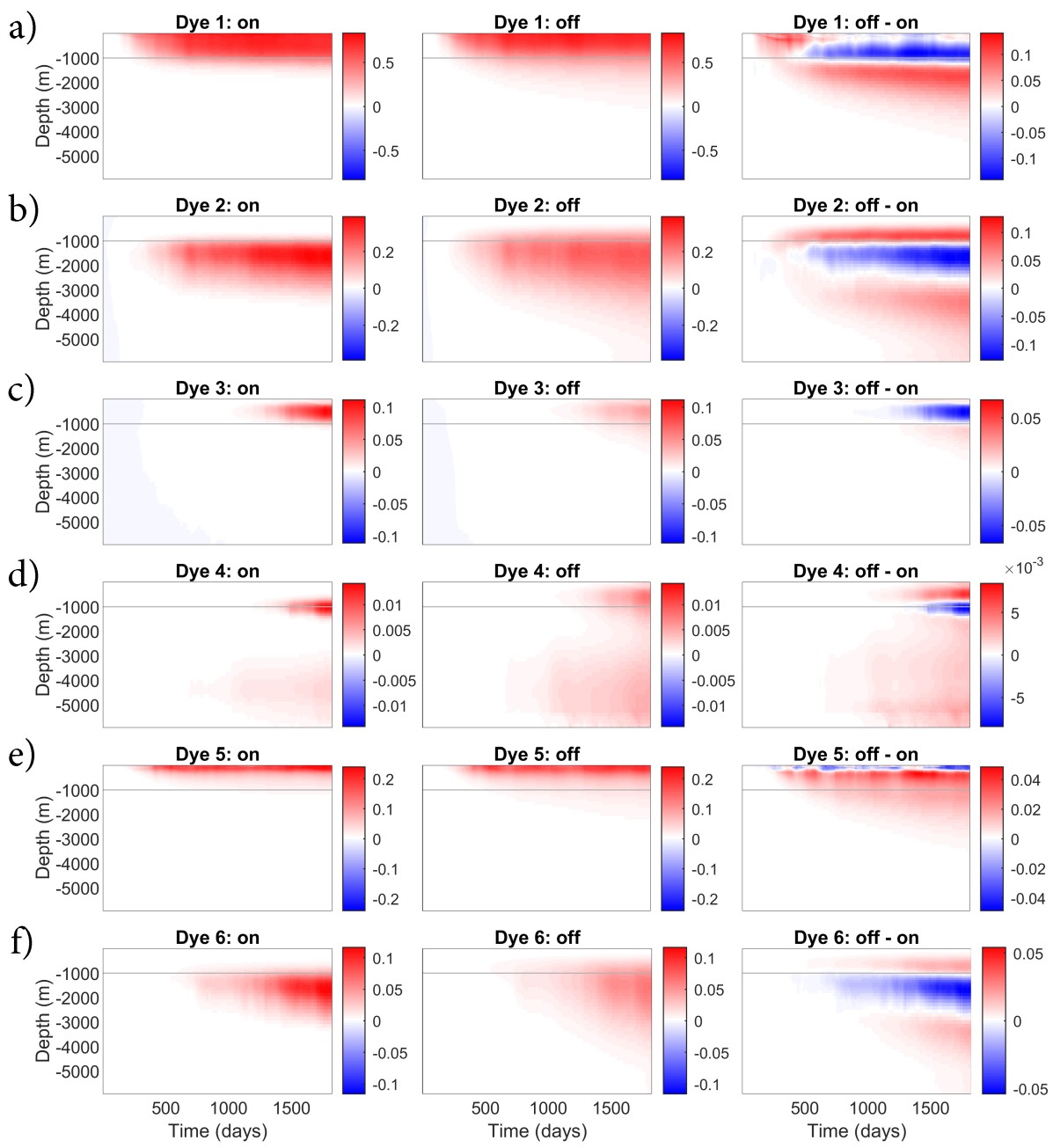

**Figure 4.** Hovmüller plots of the concentrations of the 6 dye tracers, averaged over the Zapiola Anticyclone; for the online (left column) and offline (center) simulations, and their difference (right). The 1000 m level is indicated by the light gray line.

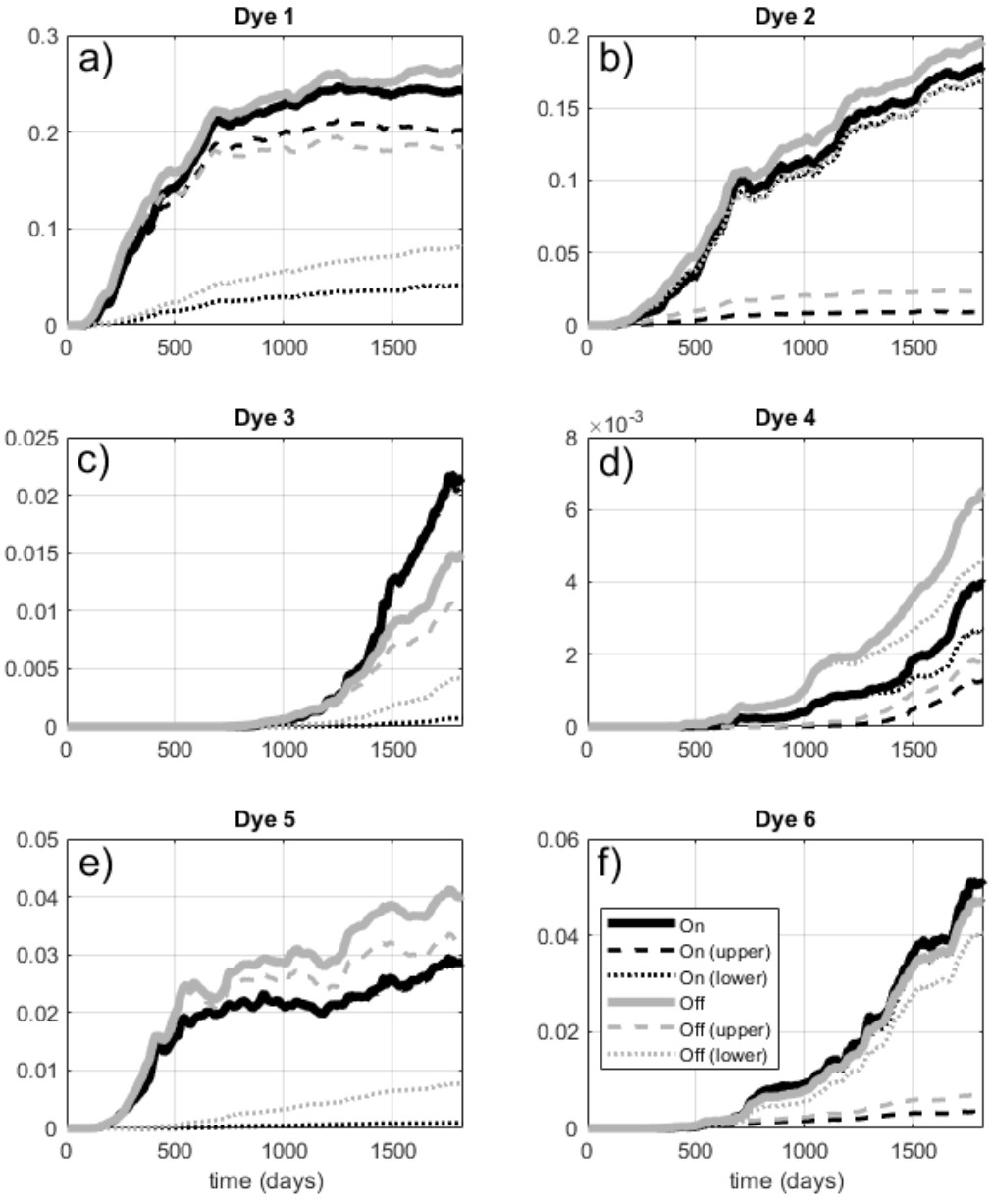

**Figure 5.** Total stock ($10^{15}$ kg) of the 6 dye tracers in the Zapiola Anticyclone, for the online (black) and offline (gray) simulations. Dashed and dotted lines are the stock above and below 1000 m, respectively.

is due to a significant vertical redistribution of tracers by vertical diffusion that depletes tracers from the upper 1000 m and increases concentrations below. Indeed, the inventory below 1000 m accounts for 30% of the column stock after 5 years.

The story is similar for dye tracer $D_2$, which is sourced at the southern boundary below 1000 m. It arrives in the ZA after 87 days in the online simulation. The stock below 1000 m (Fig. 5b, black dotted) rises quickly through the first 2 years, followed by a more gradual (linear) rise after that. Tracer concentrations at 1049 m level out at 0.20 after 3 years. The total contribution of Southern Ocean waters ($D_1 + D_2$) is about 59% at 1000 m depth. The offline simulation reproduces this behavior quite well, but displays higher inventories in the upper 1000 m that increases the overall column stock by 9% after 5 years. The vertical profiles again clearly show the impact of vertical diffusion that depletes tracers in the 1000-3000 depth range, and increases concentrations below and above (Fig. 4; second row). $D_2$ concentration at 1049 m depth levels out at 0.21, with a total Southern Ocean contribution at 1000 m of 56%.

The next-largest contribution to the ZA tracer inventory is coming from the north through Dye tracers $D_5$ (upper 1000 m) and $D_6$ (below 1000 m). It takes about 110 days for $D_5$ to arrive at the ZA, and the surface concentration saturates at about 0.18 after 550 days (as the does the upper-layer tracer stock; Fig. 5e, black dashed, overlain by solid). This suggests that the Brazil Current may contribute about 20% of the surface waters in the ZA. At 984 m, however, this fraction is still only 0.015 after 5 years, and rising, probably reflecting the strongly sheared character of the Brazil Current, and the long transit time from the northern domain boundary to the ZA. Notable concentrations of $D_6$ reach the ZA after 2 years, but trace quantities already arrive after about 270 days, having been mixed upward into the upper layer and transported southward in the Brazil Current. The offline simulations display qualitatively similar behavior, but $D_5$ inventories are significantly higher (+40%), with again significant sequestration below 1000 m. $D_6$ stock is slightly lower (-7%) than in the online simulations, despite higher values above 1000 m.

Dye tracers $D_3$ and $D_4$, released at the eastern domain boundary, take much longer to reach the ZA due, to very low westward flow velocities in the interior part of the basin. Offline stock of $D_3$ in the upper 1000 m is about 50% smaller than in the online simulation, a deficiency that can only be partly explained by sequestration below 1000 m. $D_4$ stock is 65% too high, with enhanced stock both below and above 1000 m depth.

Based on the propagation speed of the diffusion front of $D_1$ in the offline simulation (Fig. 4a, center column), we can make a rough estimate of the artificial vertical diffusivity that is introduced by the advection issue. We are able to model the depth of the diffusion front below $z_0 = -1000$ m and after $t_0 = 132$ days as $z = z_0 - \sqrt{4D(t - t_0)}$, when $D = 1.74 \cdot 10^{-2} \text{m}^2/\text{s}$. This is 2 to 3 orders of magnitude larger than typical values for background diffusivity used in ocean models.

### 3.3 The Argentine Basin test case: The Role of Temporal Averaging

The parent model is capable of producing velocity fields that have a wide range of scales of spatial and temporal variability. The shortest temporal periods are on the order of a few time steps and the longest period is the duration of the simulation. In general, higher resolution models introduce more variability on shorter length and time scales and some consideration is needed when selecting an averaging period for the transport operator diagnosis. For storage reasons, it is not practical to store snapshots of the transport operators at every time step. Conversely, representing the ocean transport with long time averages

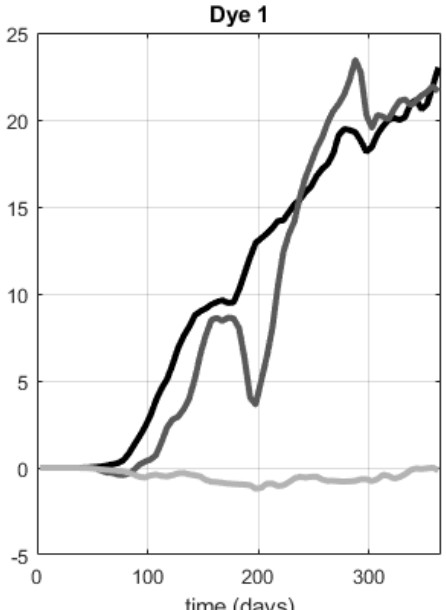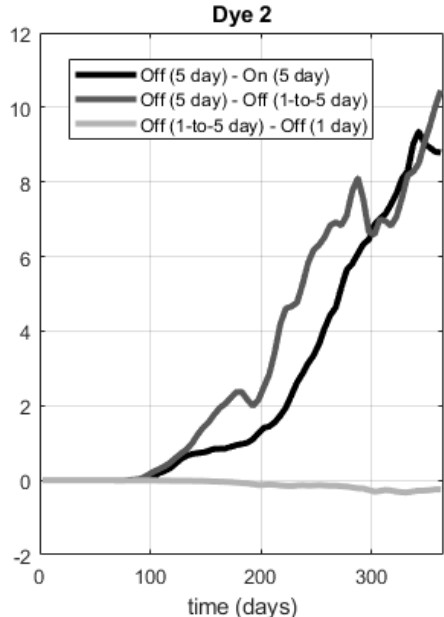

**Figure 6.** Comparison of stock ($10^{12}$ kg) of dye tracers $D_1$ and $D_2$ in the Zapiola Anticyclone for the original simulation with 5-day-averaged operators, and the alternate simulation with 1-day-averaged operators. Shown are the differences between the offline simulation using 5-day averaged operators and the corresponding online simulation (black); between the offline simulation using 5-day averaged operators, and the alternate offline simulation, with 1-day averaged operators averaged to 5 days (dark gray); and between the alternate offline simulations with 1-day averaged operators averaged to 5 days, and the 1-day averaged operators (light gray). Note that the alternative simulation is a different realization of this chaotic system, and hence diverges from the original simulation after initialization.

may exclude the effects of important variability. The choice in time averaging period for the transport operators can impact the evolution of tracers calculated in FEOTS and an appropriate balance of practicality and accuracy should be struck.

    To our knowledge, there is currently no definitive guidance on choosing the averaging period for velocity fields in an offline tracer simulation, the averaging period should be chosen so that variability in the underlying advection field is well resolved. Here, the impact of temporal averaging is investigated by comparing simulations with 1-day and 5-day averaged operators. To

that end, we ran another online simulation for which we saved 1-day averaged IRFs. This simulation was run for 105 days, producing 105 operator sets. The offline simulations were run for 365 days, hence cycling through the operator set almost 3.5 times.

    The online simulations that produced the 1-day and 5-day averaged operators were not bit-for-bit identical, as we inadvertently specified different processor counts. A positive consequence of this oversight is that we have two different realizations

of the same chaotic system, allowing us to assess how the tracer advection error compares to trajectory divergence, in terms of their impact on tracer stock. Figure 6 shows that averaging 1-day averaged operators to 5-day averages has a very small impact on the tracer stock (light gray); the impact on spatial distributions is similarly small (not shown). This shows that 5-day aver-

| Metric/Model | POP Global | FEOTS Regional | FEOTS Global (est.) |
|---|---|---|---|
| CPU-hours per sim. year | 9020 | 47 | 2115 |
| hours per sim. year | 3.7 | 7.8 | 352.5 |
| Sim. years per day | 6.5 | 3 | 0.06 |
| Cores Required | 2432 | 6 | 6 |

**Table 2.** The computational costs, model throughput, and computational hardware requirements are compared for the online parent model (POP), a regional configuration of FEOTS, and the estimated expenses for a global offline FEOTS simulation.

aged operators are sufficient to accurately reproduce tracer distributions. Comparison of the black and dark gray lines shows that the impact of trajectory divergence on tracer stock (dark gray) is of similar magnitude as the impact of the advection error
(black), at least for the first year of simulations.

### 3.4 Computational performance

One goal of FEOTS is to perform tracer calculations at a lower computational cost than the parent model. Additionally, FEOTS allows researchers to take advantage of transport operators produced by state-of-the-art climate simulations to conduct regional offline simulations. This provides flexibility in studying ocean transport phenomena and increases the value of online produced
model data while considerably reducing the computational expense for researchers solely interested in studies involving passive tracers. Here we evaluate the computational performance of a regional FEOTS configuration and compare it with the global parent model.

The total cost of using FEOTS is associated with the following steps :

1. Impulse functions are generated from the model grid,

2. An online simulation is done with POP that reads in the impulse functions and outputs IRFs averaged over the desired averaging interval (e.g., 1 day, 5 days),

3. The diagnosed IRFs are translated from gridded output to a sparse matrix format,

4. Offline passive tracer simulations are run

The first three steps are one-time costs that are necessary to generate the transport operator database. In our experience, impulse
function generation introduces a negligible cost, requiring only a few minutes to run in serial. Simulation of the passive tracers with the parent model to generate the impulse response functions requires about a factor of six more cpu-hours than when running without tracers. This is the most significant up-front cost in generating the transport operator database. Diagnosis of the sparse matrices introduces a small cost; for the parent model presented here, about 15 minutes of wall-time on a single core is needed per transport operator. The expense of the offline passive tracer simulation depends on the specific use case. Below,
we provide an example below based on the Argentine Basin test problem and a simple global simulation.

| Name | Percent Time |
|---|---|
| VerticalMixingAction | 29.73% |
| VerticalMixingPrecondition | 27.49% |
| VerticalMixing_POP_FEOTS | 18.62% |
| CalculateTendency_TracerStorage | 12.28% |
| DotProduct_POP_FEOTS | 11.19% |

**Table 3.** Flat profile of FEOTS showing the top five most expensive routines for an offline regional simulation

Table 2 summarizes a comparison of the computational expense and compute platform size requirements between POP and FEOTS. We ran the $0.3°$ ocean/sea ice configuration of E3SM-HiLAT on LANL's Institutional Computing clusters. A typical simulation with 6 dye tracers costs 9020 cpu-hrs per simulated year, using 2432 cores on 76 nodes. The throughput is 6.5 simulated years per wall-clock day. The simulation with 53 IRFs (in addition to the 6 dye tracers) typically costs 45,000 cpu-hrs, with a throughput of 1.60 simulated years per day when using 2912 cores.

In contrast, a one-year offline FEOTS simulation of the Argentine Basin problem with 6 tracers takes roughly 47 cpu-hrs, and a throughput of 3 simulated years per day on 6 cores. Table 3 shows the five most expensive routines in FEOTS and the percentage of the total wall-time spent executing those routines. The iterative treatment of vertical mixing is most expensive; a one-year simulation without vertical mixing costs 6.8 cpu-hrs, and provides a throughput of 20 simulated years per day on 6 cores. All of these routines execute sparse matrix-vector multiplication in order to compute advective and diffusive tendencies, suggesting future improvements to sparse matrix-vector multiplication in FEOTS would be beneficial.

Currently, offline global simulations at eddy resolving resolutions are slow and require a large amount of memory per core. For our global model, about 15 GB of memory is needed for the offline simulation at single precision, and 30 GB at double precision. At the time of this study, our focus was on the proving the transient simulation capabilities, which could be done at a low computational cost in regional simulations; we did not obtain direct measurements of FEOTS' runtime for global transient simulations. However, we are interested in understanding whether this methodology is potentially computationally competitive, in comparison to online tracer simulations in the parent model.

To develop an estimate for a global offline simulation with FEOTS, we assume that the cpu-hours scale linearly with the number of grid cells. This is reasonable since the majority of FEOTS' runtime is spent in sparse matrix-vector multiplication, which scales linearly with the number of rows. The Argentine Basin simulation has $1.02 \times 10^6$ grid cells, whereas a global configuration about 45 times more grid cells. Thus, an offline simulation with FEOTS is expected to cost about 2115 cpu-hours per simulation year in the current version, 77% less cpu-hours than the parent model. However, the expected throughput for an offline global simulation is estimated as 1 model year for every 15 days. Although FEOTS is currently expected to take longer to produce global transient simulations, it is probable that exposing parallelism in FEOTS will provide comparable run-times to the parent model at a reduced computational expense. Though this is not a definitive result, it is a reasonable estimate that motivates future work in exposing parallelism in FEOTS.

## 4  Discussion

Based on this work, we posit that offline tracer simulations provide a viable modeling capability at a significantly reduced computational expense compared to online models. With the reduction in computational expense, however, we have shown that the offline simulations can produce tracer distributions consistent with a more diffusive ocean circulation than online models. A potential approach to mitigate this problem, when using nonlinear flux-limiting advection schemes, is to replace the impulse fields with smoother basis functions, as discussed by Khatiwala et al. (2005).

When using impulse fields with sharp discontinuities, the 3rd Order flux-limited Lax-Wendroff advection scheme reduces effectively to a first order upwind scheme. By leveraging smoother basis functions for the impulse fields with this scheme, a less diffusive advection operator can be diagnosed. Using a smoother basis function, however, is expected to introduce additional complications for diagnosing transport operators. In general, the process of diagnosing the transport operators can be thought of as a matrix projection problem,

$$\mathbb{A}\mathbb{F} = \mathbb{R} \tag{22}$$

where $\mathbb{A}$ is the transport operator we want to diagnose, $\mathbb{F}$ is a matrix whose columns are the impulse fields, and $\mathbb{R}$ is a matrix whose columns are the impulse response fields. The advection operator is obtained by multiplying (22) by the right-inverse of $\mathbb{F}$

$$\mathbb{A} = \mathbb{R}\mathbb{F}^{-1} \tag{23}$$

Ideally, the basis function we choose should be smooth enough to retain higher order terms in the 3rd Order flux-limited Lax-Wendroff advection scheme. For computational purposes, the basis functions would ideally be mutually orthonormal so that the inverse is easy to calculate,

$$\mathbb{F}^{-1} = \mathbb{F}^{T} \tag{24}$$

This approach, along with experimentation with other advection schemes, is planned for future work.

With regards to performance, FEOTS can provide offline tracer modeling capabilities at significantly reduced computational expense (cpu-hours) and with practical runtimes for regional simulations. Projected computational resource requirements (cpu-hours) were shown to be about 77% less than the online parent model. Estimated runtimes for global simulations, however, indicate that parallelism FEOTS needs to be exposed before it is practical for this use case. Addressing the slow runtime for global offline simulations is critical for working towards a framework that allows for the calculation of steady state solutions within a practical amount of time. Currently, our plan for reducing runtime is to leverage open-source toolkits, like PSBLAS (Filippone and Colajanni, 2000), to handle sparse matrix operations in parallel

## 5  Conclusions

In this paper we introduced the Fast Equilibration of Ocean Tracers Software (FEOTS), which is an end-to-end set of tools to efficiently calculate tracer distributions on a global or regional sub-domain. Key features currently implemented in FEOTS and

discussed in this paper include impulse field generation through graph coloring, impulse response function translation into a sparse matrix representation of transport operators (advection), regional sub-domain transport operator generation from global transport operators, and offline forward simulation with the diagnosed operators. These are key components for calculating equilibrium tracer fields, which we aim to implement in future releases. In our example problem we diagnose transport operators from a global eddy-permitting configuration of the POP ocean model, and calculate the distribution of passive dye tracers in the Argentine Basin for a 5-year period. The offline simulations are shown to produce more diffuse tracer distributions than the online parent model simulations This demonstration shows the feasibility of this approach, while at the same time highlighting the challenges of the Impulse Response Functions approach.

*Code and data availability.* The current version of FEOTS (v0.0.0) is available from the project website: https://github.com/FluidNumerics/FEOTS/ under the 3-Clause BSD licence. The exact version of the model used to produce the results used in this paper is archived on Zenodo (https://doi.org/10.5281/zenodo.5576912). The input data and scripts to run the model and produce the plots for all the simulations presented in this paper are also archived on Zenodo (DOI: 10.5281/zenodo.6250938). A codelab tutorial to walk-through running the Argentine Basin simulations is available online at https://fluidnumerics.github.io/FEOTS/codelabs/feots-on-google-cloud/#0.

*Author contributions.* Schoonover and Weijer wrote the manuscript. Schoonover conceived, developed and wrote the FEOTS code, and performed the error analysis; Weijer performed online and offline tracer simulations and analysis; and Zhang contributed to the development and testing of FEOTS.

*Competing interests.* The authors declare that no competing interests are present.

*Disclaimer.* TEXT

*Acknowledgements.* We are grateful to Keith Lindsay (NCAR), and Ann Bardin and François Primeau (UCI), for sharing code, and for useful discussions. This research was supported by the Los Alamos National Laboratory (LANL) through its Center for Space and Earth Science (CSES). CSES is funded by LANL's Laboratory Directed Research and Development (LDRD) program under project number 20210528CR. Joseph Schoonover and Jiaxu Zhang acknowledge support from LANL's Center for Nonlinear Studies (CNLS). Computations were performed on LANL's Institutional Computing platforms, and on the Darwin cluster. This research was supported by the Regional and Global Model Analysis (RGMA) component of the Earth and Environmental System Modeling (EESM) program of the U.S. Department of Energy's Office of Science, as contribution to the HiLAT-RASM project. We thank Ann Bardin and an anonymous reviewer for constructive reviews of this paper.

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
