# Peer review of "FEOTS v0.0.0: A New Offline Code for the Fast Equilibration of Tracers in the Ocean"

_Geoscientific Model Development, 2022_

## Referee Comment (RC1)

Review of Shoonover et al preprint titled "FEOTS: A New Offline Code for the Fast Equilibrium of Tracers in the Ocean"

General Comments:
This well-written manuscript explores the challenge of applying transport matrix technology to a higher-resolution parent model (nominal 0.3-degree, 100 depth levels), where there is even more to be gained than in the typical 1-degree model. There is a greatly enhanced technique for generating the IRF stencils such that a minimal number of tracers are needed to capture the IRF results. The use of a set of regional transport matrices to analyze the circulation in the Argentine Basin highlights the use of the transport matrix technique to develop a regional model for in-depth analysis of a particular area of the ocean. This facilitates gaining meaningful insights at relatively low computer cost.

While this is a useful exploration, in the example, the transport matrices are simply run forward for a few years; there is no "fast equilibrium" solution attempted. Therefore, the paper should be retitled to better describe its actual content, and what was accomplished.

Specific Comments:

@ line 50
Rephrase this to reflect what is accomplished, and to distinguish it from future plans.

@ 62
Convergence rates? Perhaps you mean volume transport rates? (So as not to be confused with the convergence rates to an equilibrium solution.)

@ 93 Section Graph Coloring approach to Operator Diagnosis
This would be easier to comprehend after having read Section 2.4 Parent Model, where the advection scheme is discussed, and the motivation for the graph coloring approach becomes clear.

@ 125
 State which advection scheme (Table 1) was actually used in the offline transport matrix model for the example in this paper.

@ 150 and forward
Somewhere (Section 2.4?) please indicate the timestep used, and the relationship to the CFL for both the parent and the transport matrix model.

@ 164 and forward
From this it is assumed that there is NO horizontal diffusion term, as none is generated by the online parent model. Please restate the offline equation (eq. 10) with the missing horizontal diffusion term, and clarify which of the options in Table 1 was used.

@175
A 63-year spin-up seems short for examining deep water masses; the basin extends to 6000m. A comparison with observational temperature, salinity, and flow statistics as was done in Weijer et al, 2020 would be a good addition.

@177
Is the data volume given for IRF-output and transport operators for the entire ocean, or for the region to be studied?

@ 178 and forward
Clearly state when you are running the global model versus the regional model.

@ 200 and forward
This is a clear and useful discussion of the constant preservation issue, which is frequently overlooked until strange results show up. An additional useful criterion is the rate at which total tracer quantity is gained or lost. This will impact the ability to get a physically meaningful equilibrium solution.

@ 203
The use of single precision for the parent model will limit the usefulness of the results. It also makes it difficult to distinguish the sources of errors in the analysis.

@212 this paragraph
Clarify that the error data is from running the global or regional version of the offline model. There appears to be an assumption that these errors are not present in the parent dynamic model. It would be instructive to run this test on the dynamic model, in order to compare the error level inherent in the parent dynamic model.

@ 222
…"with and without mixing": vertical mixing? Please clarify.

@233
How does the 1000m division relate to the mixed-layer depth of the model?
In Figures 4 and 5, Dye 4, especially, raises the question about the depth of the mixed layer in this vicinity.

Figure 4:
The scales on the figures are not all the same. This creates the visual impression that the small contributions are more important than they are, compared to the major contributors.

Figure 5:
Interesting set of graphs.

Again, the scales on the graphs are very different,  creating a visual that is out of proportion for the more slowly arriving contributors.

@340
Supply a reference for the statement : …"reduces to a first order upwind scheme".

@361
Need to add finding the equilibrium solution  under future work.

@368
Your conclusion that the analysis "highlights the limitations of the IRF approach", considering the limitations of this particular implementation,  might be more appropriately stated as "highlights the challenges of the IRF approach".

---

## Referee Comment (RC2)

**FEOTS: A New Offline Code for the Fast Equilibration of Tracers in the Ocean**

Joseph Schoonover[1], Wilbert Weijer[2], and Jiaxu Zhang[3,4]

[1]Fluid Numerics LLC., Boulder, CO 80304
[2]Los Alamos National Laboratory, Los Alamos, NM 87545
[3]Cooperative Institute for Climate, Ocean, and Ecosystem Studies, University of Washington, Seattle, WA 98105
[4]NOAA/Pacific Marine Environmental Laboratory, Seattle, WA 98115

**Correspondence:** Joseph Schoonover (joe@fluidnumerics.com)

**Abstract.** In this paper we introduce a new software framework for the offline calculation of tracer transport in the ocean. The Fast Equilibration of Ocean Tracers Software (FEOTS) is an end-to-end set of tools to efficiently calculate tracer distributions on a global or regional sub-domain using transport operators diagnosed from a comprehensive ocean model. To the best of our knowledge, this is the first application of a Transport Matrix Model to an eddying ocean state. We demonstrate the capabilities of FEOTS in an application focused on the Argentine Basin, where intense eddy activity and the Zapiola Anticyclone lead to strong mixing of water masses. The demonstration clearly shows the benefits of the approach, while at the same time highlighting the limitations of the Impulse Response Functions approach in capturing tracer transports by a non-linear advection scheme. Our future work will focus on improving the computational efficiency of the code to reduce time-to-solution, and on using different basis functions to better represent non-linear advection operators.

You don't say the nature of these benefits: is your method faster? More accurate? Also, I don't know if "clearly shows" is the best phrasing, because it is not determined whether you approach is more accurate than the online approach (it seems more diffusive) and in terms of time, as far as I understood, the comparison is done on estimated costs and not actually running the online vs offline head to head.

**1 Introduction**

Many oceanographic research problems involve the transport and distribution of tracers that do not feed back on the ocean dynamics. Examples of such problems are the diagnostic tracking of water masses using passive tracers (e.g., Dukhovskoy et al., 2016; Zhang et al., 2021); validating the use of isotopes or grain size distributions in marine sedimentary records to infer past ocean circulation changes (e.g., Jahn et al., 2015; Zhang et al., 2017; Gu et al., 2019; Missiaen et al., 2020); assessing anthropogenic carbon uptake by the ocean (e.g., Sarmiento et al., 1992; Khatiwala et al., 2009; Wang et al., 2012); studying the evolution of marine biogeochemical systems (e.g., Séférian et al., 2020); or tracking the fate of microplastics in the ocean (e.g., Mountford and Morales Maqueda, 2019). In many cases, the transport of these tracers is dominated by mesoscale processes like eddies. Typical ocean climate models use grids that are too coarse to explicitly resolve these processes and rely on parameterizations to simulate their impact on tracer fields; but it has become clear that these eddy-parameterized models fail to reproduce some critical aspects of the real ocean (e.g., Lozier, 2010). What is more, the ocean also contains *dynamical* features that rely on mesoscale eddies, and which cannot be reproduced by low-resolution models.

A case in point is the Argentine Basin. It is among the most turbulent regions in the World Ocean (Fu and Smith, 1996), mostly on account of the confluence of two western boundary currents, the Brazil and Malvinas Currents (e.g., Garzoli, 1993).

[Figure]

A seamount in the center of the basin is associated with a local minimum in eddy kinetic energy (Fu and Smith, 1996), but

25   is surrounded by a very strong barotropic vortex, the Zapiola Anticyclone (ZA; Saunders and King, 1995; de Miranda et al.,
1999). This anticyclone is understood to be driven by the intense eddy field (Dewar, 1998), and has been shown to inhibit
exchanges between the interior of the ZA and its surroundings (Weijer et al., 2015, 2020). Strikingly, the Argentine Basin is
also a main conduit of water mass exchange between the Atlantic and Southern Oceans (e.g., Jullion et al., 2010), so eddy-
driven mixing of water masses in the Argentine Basin may have implications for the global thermohaline circulation. It is clear

30   that studying tracer transport and mixing in the Argentine Basin requires an ocean model that resolves the ocean's mesoscale;
not just to accurately represent the transports by narrow boundary currents and the turbulent eddy field, but also to generate
the ZA in the first place. Second, the equilibration of tracers at deeper levels may take many decades or centuries, requiring
a model capability that can be run for such long times, or can determine equilibria directly using iterative solvers. Third, this
problem is ideally addressed using a representation of the global ocean circulation; not only so that the ocean circulation in

35   the region of interest is fully consistent with large-scale ocean [I think you should spell out what the state of the art approach is and in what way your method improves it. Your narrative is a little too vague.]
regional problem would require a unique model configuration.

The cost and technical challenges of running global and fully-dynamic ocean models make it expensive and difficult at best,
but often impossible, to address problems like these. It is obvious that there is a need for simple and efficient tools that solve
tracer transport and distribution problems, without the need to explicitly simulate ocean dynamics. In the past decade or so,

40   several Transport Matrix Models (TMMs) have been developed that allow for the simulation of passive tracers in a stand-
alone (offline) code, using transport operators that have been diagnosed from comprehensive ocean models (Primeau, 2005;
Khatiwala et al., 2005; Khatiwala, 2007; Bardin et al., 2014). In particular, Khatiwala et al. (2005) pioneered the approach of
empirically estimating an ocean model's transport processes by diagnosing the action of the transport (advection and diffusion)
operators on simple basis functions like impulse fields. The resulting Impulse Response Functi[Could you expand on the current state of the art for the IRF approach and say more on what the differences with your work are?]on in t

45   [Although above you said that climate models use parametrizations to describe the presence of eddies, can you clarify?]
tationa

Simulations are now routinely being produced at eddy permitting and eddy resolving scales. To the best of our knowledge, the capability introduced in this paper is the first TMM developed for and applied to eddying ocean states, with transport operators diagnosed from a global ocean model with nominal resolution of 0.3° and 100 vertical levels ($\sim 10^8$ degrees of freedom).

50   Our computational framework is embodied in the Fast Equilibration of Ocean Tracers Software (FEOTS; https://github.com/
LANL/FEOTS). FEOTS generalizes the methodology from Bardin et al. (2014) by providing an end-to-end set of tools that
i) use an advanced optimization algorithm to generate an optimal set of impulse functions, given the grid layout and operator
stencils for the parent model; ii) transform the resulting IRFs obtained from the parent model to transport operators; iii) set
up regional or global tracer problems, with different types of tracers; iv) run forward simulations of the offline tracer model;

55   and v) use a Newton-Krylov solver to determine steady tracer distributions. The framework is applied here to the Parallel
Ocean Program (POP; Smith et al., 2010); but the design is flexible enough to be generally applicable –in particular to the new
generation of ocean models with unstructured grids, like the Ocean Model for Prediction Across Scales (MPAS-Ocean, Ringler
et al., 2013).

[Figure]

You may want to rephrase this and just say what the magnitude of the error is (like 10^-something)

In this paper we present validation and verification results for the FEOTS offline tracer solver in a regional forward simulation configuration focused on the Argentine Basin. Specifically, we will show that uniform tracer fields are preserved within 0.01% after five years of offline model integration and implemented time integration schemes exhibit the expected convergence rates. Additionally, we will present a comparison of the offline regional tracer simulation using five-day averaged transport operators with an online tracer simulation from the parent model.

**2 Methodology**

What application?

Our work builds on the methodologies of Bardin et al. (2014) to create FEOTS. This application is used to aid in the capturing of transport operators from ocean model simulations and in the execution of offline regional transient tracer simulations. In section 2.1 we present the governing equations for a non-interacting passive-tracer system and outline the methodology for capturing transport operators from a comprehensive ocean model in 2.2. The offline forward stepping algorithm and treatment of vertical mixing is presented in section 2.3. In section 2.4 we discuss the parent model POP, while the Argentine Basin test problem is defined in 2.5.

**2.1 Governing Equations**

We model a passive dye tracer as a concentration field that is subjected to advection and diffusion,

$$\frac{\partial c}{\partial t} + \nabla \cdot (\boldsymbol{u}c - \mathbb{K}\nabla c) = 0 \tag{1}$$

where $c$ is the tracer concentration, $t$ is time, $\boldsymbol{u}$ is the ocean velocity field, and $\mathbb{K}$ is the diffusivity tensor that models unresolved eddy activity. The initial and boundary conditions are set to be

$$c(t = 0) = c_0(z, \theta, \phi) \tag{2}$$

and

$$c = c_b(z, \theta, \phi, t), \tag{3}$$

What discretization method are you using? where are the unknowns located on the mesh? what type of cells are you using? The section on the Parent Model should be moved up so this sentence is put into context for the reader.

where $z$ is depth (measured positive downward), $\theta$ is longitude, and $\phi$ is latitude. The semi-discrete form of (1) can be written as:

$$\boldsymbol{c}_t + (A + D_h + D_v)\boldsymbol{c} = 0 \tag{4}$$

where $\boldsymbol{c}$ is a vector of the discrete[...] d $D_v$ is the vertical diffusion matrix.

Why is your offline approach better than just discretizing the operators and coding them up? What is the accuracy of this method? Has it been tested against toy problems for which there was an analytical solution? If so, proper references should be cited here. If not, at least a heuristical discussion on accuracy should be addedd.

The transport operators in equation (4) can be diagnosed empirically from a comprehensive ocean model, the "parent model", using the methodology used by Bardin et al. (2014) and pioneered by Khatiwala et al. (2005). In the parent model, passive tracers are initialized to a set of impulse functions $[\boldsymbol{c}^{(i)}]_{i=1}^{M}$ at the beginning of each time step, where $M$ is the number of

[Figure]

[Figure]

> Please define in mathematical terms what you mean by "tendency of the impulse function".

impulse functions. At the end of each time step, the tendency of the impulse function, called the impulse response function (IRF) is captured by diagnosing the advective tendency,

$$\boldsymbol{R}^{(i,n)} = A^{(n)}\boldsymbol{c}^{(i)} \qquad (5)$$

> Is Eq. (14) a better way to write this?
> If A is the operator you are trying to estimate and R is known data, you should probably just write it as in Eq. (14).

where $\boldsymbol{R}^{(i,n)}$ is the impulse response function at time level $n$ associated with impulse function $i$, $A^{(n)}$ is the transport matrix at time level $n$, and $\boldsymbol{c}^{(i)}$ is the $i^{th}$ impulse function. The IRFs are averaged over a certain averaging period and written out to file.

**2.2 Graph Coloring approach to Operator Diagnosis**

The purpose of capturing the impulse response functions is to diagnose sparse matrices that are consistent with the advection discretization in the parent model. To illustrate this procedure, suppose that the parent model has $N$ grid cells and that $N$ impulse fields are set as the Kronecker delta functions:

$$[\boldsymbol{c}^{(i)}]_{i=1}^N = [\delta_{j,i}]_{i=1}^N \qquad (6)$$

In this setup, impulse function $i$ is zero at all grid cells except for grid cell $i$ where the impulse function has a value of one. Application of the transport matrix to each of the $\boldsymbol{c}^{(i)}$ returns column $i$ of $A$,

> Does this procedure depend on the time step? In the fully discretized system the entries of A are multiplied by delta t (time-step) I believe. Do you need to estimate again A any time the delta t for your simulation changes?

$$Ac^{(i)} = \sum_{j=1}^N A_{m,j}\delta_{j,i} = A_{m,i} \quad \text{for } m = 1,2,3,...,N$$

While using a set of Kronecker delta functions will completely diagnose all of the elements of the transport matrix, this strategy is computationally expensive. For each time step, this strategy requires computing the advective tendency for $N$ tracer fields, where $N$ is the number of grid cells. For example, coarse resolution model at $\mathbb{O}(1°)$ resolution have roughly $10^6$ grid cells. Storing $10^6$ impulse and impulse response functions would require approximately 3 TB of memory at double precision.

To reduce the number of required impulse functions to fully diagnose the transport matrices, we can take advantage of the fact that the advection scheme results in a sparse matrix. Equivalently, the domain of influence of the advection operator is limited to nearby grid cells. The parent model employed in Bardin et al. (2014) used a third order upwind scheme, where the impulse response is guaranteed to extend no further than two grid-cells in each spatial dimension. Because of this, the authors used a set of 125 tracer fields,

> Is this the main difference between your approach and Bardin 2014?
> Why 125 tracers and not another number?

$$c(i,j,k;i_0,j_0,k_0) = \delta_{i0,i(mod5)}\delta_{j0,j(mod5)}\delta_{k0,k(mod5)} \quad \text{for } i_0 = 1,...,5; j_0 = 1,...,5; k_0 = 1,...,5 \qquad (8)$$

FEOTS offers a unique capability to generate a minimal set of impulse functions by posing the problem as a graph coloring problem. A graph $G(V,E)$ is defined by a set of vertices $V$ and edges $E$ that connect the vertices. Two vertices connected by an edge are said to be adjacent. A valid graph coloring of $G(V,E)$ assigns colors to each vertex so that no two adjacent vertices have the same color. To calculate impulse functions that can be used to diagnose transport operators, FEOTS offers functionality to express a POP mesh and an advection stencil into an equivalent graph that is colored with a Greedy algorithm.

[Figure]

[Figure]

| Method | $c^*$ |
|---|---|
| Forward Euler | $c^n$ |
| Adams-Bashforth 2nd Order | $\frac{3c^n - c^{n-1}}{2}$ |
| Adams-Bashforth 3rd Order | $\frac{23c^n - 16c^{n-1} + 5c^{n-2}}{12}$ |

**Table 1.** Optional values for $c^*$ in Eq. (10), based on the choice in time integration scheme.

This formulation has the benefit that it can be generalized to parent models based on unstructured grids and it takes into account irregular boundaries from variable bathymetry.

In FEOTS, graph vertices $V$ correspond to each ocean grid cell, centered on tracer points, in the POP mesh. Two vertices are adjacent if their impulse response functions overlap. Because a valid coloring results in adjacent vertices having distinct colors, vertices with the same color can safely be assigned to the same impulse function. Consequently, the chromatic number of the graph corresponds to the number of impulse functions used for model diagnosis. For this work, the parent model uses a $0.3°$ periodic tripole mesh and the 3rd order flux-limited Lax-Wendroff advection scheme. This approach results in 53 impulse functions required to uniquely diagnose the transport operators.

**2.3 Time integration**

Forward integration of the offline tracer model uses Backward Euler for vertical mixing and can use Forward Euler, Adams-Bashforth $2^{nd}$ Order, or Adams-Bashforth $3^{rd}$ Order for transport. As in Bardin et al. (2014), we forward step an equation for the volume anomaly using a forward Euler method. Volume anomalies arise due to divergence in the transport field at the upper-most z-level that are associated with fluctuations of the free-surface.

In general, the time integration scheme can be written as

> [Who is v? it was not in Eq. (4)]

$$v^{n+1} = v^n + \Delta t \mathbb{A}^n i \tag{9}$$

> [How are the volume anomalies discretized? if one wanted to reproduce what you do, how do they deal with the volume anomalies?]

$$(\mathbb{I} + \mathbb{V}^{n+1} + \mathbb{D}_v)c^{n+1} = (\mathbb{I} + \mathbb{V}^n)c^n + \Delta t(\mathbb{A} + \mathbb{D}_h)c^* \tag{10}$$

[revised manuscript text omitted]

**3 Results**

In this section, we present results of an offline regional study focused on the Argentine Basin. We first present verification of
195 the constant preservation property (subsection 3.1). Then we present a comparison of the tracer simulation results in the parent model (online) and the offline model using 5-day averaged transport operators (subsection 3.2), followed by a comparison between 1-day and 5-day averaged transport operators (subsection 3.3). Finally, we will discuss the computational performance of the code (subsection 3.4) on the systems where our simulations were conducted.

**3.1 Constant Preservation**

The FV discretization should be mentioned when you bring up the semi-discrete model

200 The parent model uses a finite volume discretization that guarantees the preservation of constant tracer fields. To verify that FEOTS accurately diagnoses transport operators that are representative of the parent model, our first simulation involves verifying that a constant tracer field remains constant under the action of the diagnosed transport operators.

[Figure]

[Figure]

**Figure 1.** Vertical profile of the max volume anomaly after 5 days (gray) and after 5 years (black). In the exact form of the equations, the volume anomaly should only exist in the first vertical layer at the ocean surface. Non-zero values in the volume beneath the surface layer arise due to round-off errors.

> But is it a physically reasonable choice? Also, the way you test accuracy is potentially hindered by having a smaller number of significant digits.

In all of our simulations, we have opted to use single precision arithmetic and have enabled aggressive compiler optimizations (compiler option *-Ofast* with GCC 9.2.0). These choices were made to minimize data storage costs for the transport operators and post-processing output and to optimize the time-to-solution for the offline simulations. Although analytically, we expect that the FEOTS algorithm should preserve constant tracer fields, errors from floating point arithmetic are expected to be the main source of constant preservation errors.

To better understand the spatial distribution and the relative impact of round-off errors, we configure a simulation where the initial tracer field is set to $s_0 = 1$ and there is no external source or sink of tracer. Since $\nabla s_0 = 0$ and $\nabla \cdot \boldsymbol{u} = 0$ under the discretizations used in the parent model, we expect that $s_t = 0$ and $s = s_0 = 1$ for all time. For this experiment, we use the 5-day average operators and forward step the model for 5 years to cycle through the 365 diagnosed transport operators once.

Figure 1 shows the maximum volume anomaly in the domain as a function of depth after five days and five years of integration. Analytically, the volume anomaly is expected to be zero in all cells except the top-most layer. At the surface layer, fluctuations in the free surface height are associated with non-zero fluid divergences that contribute to changes in the fluid volume. Beneath the surface layer, the fluid velocity field is expected to be divergence free. In general, larger errors in the volume anomaly are observed above 1000 m depth. After five days, errors in the deep ocean are $\mathbf{O}(10^{-5})$ and after five years, the deep ocean volume anomaly errors have grown by an order of magnitude to $\mathbf{O}(10^{-4})$. Larger errors are observed above 1000 m, reaching $\mathbf{O}(10^{-3})$ after 5 years. Note that the volume anomaly field is identical for all choices in time integrator for the dye tracer and is independent of vertical mixing.

> Do you consider these to be small errors? for single precision, machine precision is around 10^-8 and here it looks like you are several orders of magnitude above that. Can you comment?

Errors in the volume anomaly lead to spurious values [...]on of the tracer concentration from its initial uniform value is erroneous. Figure 2 shows the max error in the dye tracer as a function of depth after five days and five years of integration, with and without mixing. After five years of integration, the maximum relative error with mixing is about $0.05\%$ and without mixing is about $0.01\%$. At depth, the errors in the tracer are

[Figure]

It would be interesting to see how these plots compare to doing the same test for the online method

[revised manuscript text omitted]

> Can you comment on how much the nature of the problem affects the averaging methodology?

[Figure]

[Figure]

| Metric/Model | POP | FEOTS Regional | FEOTS Global (est.) |
|---|---|---|---|
| CPU-hours per sim. year | 9020 | 47 | 2115 |
| Sim. years per day | 6.5 | 3 | 0.06 |
| Cores Required | 2432 | 6 | 6 |

**Table 2.** The computational costs, model throughput, and computational hardware requirements are compared for the online parent model (POP), a regional configuration of FEOTS, and the estimated expenses for a global offline FEOTS simulation.

**3.4 Computational performance**

> Then maybe this paragraph should be the first one you discuss in your results section.

290   The primary goal of FEOTS is to perform tracer calculations at significantly lower cost than the parent model. Additionally, FEOTS allows researchers to take advantage of transport operators produced by state-of-the-art climate simulations to conduct regional offline simulations. This provides flexibility in studying ocean transport phenomena and increases the value of online produced model data while considerably reducing the computational expense for researcher solely interested in studies involving passive tracers. Here we evaluate the computational performance of a regional FEOTS configuration and compare it with

295   the global parent model.

The total cost of using FEOTS is associated with the following steps :

1. Impulse functions are generated from the model grid,

2. The impulse functions are then repeatedly passed through the passive tracer components of POP while running an online simulation,

> Can you clarify how the averaging processes works? For instance, considering your 1 day and 5 day averaged operators, how many runs did you do for each of them?

300   3. The diagnosed IRFs are translated from gridded output to a sparse matrix format,

4. Offline passive tracer simulations are run

The first three steps are one-time costs that are necessary to generate the transport operator database. In our experience, Impulse function generation introduces a negligible cost, requiring only a few minutes to run in serial. Simulation of the passive tracers with the parent model to generate the impulse response functions requires about a factor of six more cpu-hours than when

[revised manuscript text omitted]

---

## Author Comment (AC1)

- @ line 50 Rephrase this to reflect what is accomplished, and to distinguish it from future plans.
  - We've rewrote the paragraph to be more specific about the differences between FEOTS and other capabilities, and to be clear about the current state of FEOTS and what we plan to do in the future.

- @ 62 Convergence rates? Perhaps you mean volume transport rates? (So as not to be confused with the convergence rates to an equilibrium solution.)
  - Here we were talking about the error convergence rate of the time integration scheme. For example, the Forward Euler method exhibits linear convergence with the time step size. When we were drafting this paper, we initially had plans to look into convergence rates. We ran into a few issues when looking at this and opted to leave this out of the paper. To avoid any confusion, we've removed the mention of convergence rates at this line.

- @ 93 Section Graph Coloring approach to Operator Diagnosis This would be easier to comprehend after having read Section 2.4 Parent Model, where the advection scheme is discussed, and the motivation for the graph coloring approach becomes clear.
  - We have moved the previously named "section 2.4" to occur immediately before this section; Reviewer #2 also made this request.

- @ 125 State which advection scheme (Table 1) was actually used in the offline transport matrix model for the example in this paper.
  - We have added the following to the caption on Table 1

    *3rd Order Adams-Bashforth is used for the Argentine Basin test problem presented in this paper.*

  - We have added the following to the end of the "Time Integration" section

    *For the results presented in this paper, we use the 3rd Order Adams-Bashforth time integrator and the conjugate gradient solver is stopped when the residual magnitude, relative to the initial solution guess magnitude, is less than $10^{-6}$.*

- @ 150 and forward Somewhere (Section 2.4?) please indicate the timestep used, and the relationship to the CFL for both the parent and the transport matrix model.
  - We diagnosed the CFL values from the standard POP diagnostic output, and calculated the maximum CFL by performing an eigenvalue analysis on a transport matrix. The latter procedure was performed using matlab on a regional operator for the Argentine Basin, as memory limitations prevented us from doing so on a global operator. The dominant eigenvalue indicated a CFL value of 0.1 for a 15-minute time step.

○ We added the following text to the "Parent Model" section:

*With a time step of 7 minutes, the model typically yields maximum CFL values of $\textbf{O}(10^{-1})$ or smaller.*

○ We added the following text to the "Time Integration" section:

*We use a 15 minute time step, and a typical maximum CFL value, obtained by eigenvalue analysis of the transport operators, is $\mathcal{O}(0.1)$.*

- @ 164 and forward From this it is assumed that there is NO horizontal diffusion term, as none is generated by the online parent model. Please restate the offline equation (eq. 10) with the missing horizontal diffusion term, and clarify which of the options in Table 1 was used.
  ○ You are correct; there is no explicitly implemented horizontal diffusion term. Lateral tracer diffusion comes from the diffusive nature of the Flux-Limited Lax Wendroff scheme when it is applied to impulse fields. We have added the following statement immediately after Equation (4):

  *Since we do not enable explicit lateral tracer diffusion in the parent model in this study, all elements of $D\_h$ are zero; we do not explicitly diffuse tracer laterally.*

- @175 A 63-year spin-up seems short for examining deep water masses; the basin extends to 6000m. A comparison with observational temperature, salinity, and flow statistics as was done in Weijer et al, 2020 would be a good addition.

  ○ In this paper, we are not concerned with the deep water masses per se, but rather with the effect of the circulation on tracer distributions. After 63 years, the primary circulation features (boundary currents, the eddy field, the Zapiola Anticyclone) are well established. We added the following text to clarify:

  *Although the model was run for 186 years, we diagnosed the transport operators for the 5-year period starting at simulation year 64. Even though 63 years of spin-up is not sufficient to fully equilibrate the stratification in the deep ocean, the main circulation features (e.g., boundary currents, the eddy field, the Zapiola Anticyclone) are well established by then, making this an appropriate data set to demonstrate the capability of FEOTS. We refer to \cite{Weijer2020a} for evaluation of the hydrography and circulation in the Argentine Basin in a companion simulation.*

- @177 Is the data volume given for IRF-output and transport operators for the entire ocean, or for the region to be studied?

- ○ The parent model is a global climate model. The data volume is for the whole parent grid (global). We use this database as a source to generate either global or regional transport operators from. The paragraph has been moved to the "Graph Coloring" section and has been revised to read:

  *For our test problems, we diagnosed the 5-day averaged IRFs and vertical diffusivities for the 5-year analysis period of the parent model. We repeated the simulation for 105 days, diagnosing 1-day averaged IRFs. With this methodology and the 7-minute time step, the one-day averaged operators are each an average of 1440 IRF snapshots and the five-day averaged operators are each an average of 7200 IRF snapshots. The data volume of the global parent model five years' worth of 5-day averaged operators (365 IRFs and diffusivities) is about 9 TB. Once transformed to transport operators, the data volume is 4 TB.*

- @ 178 and forward: Clearly state when you are running the global model versus the regional model.
  - ○ We've reviewed the manuscript and made adjustments following this recommendation

- @ 200 and forward: This is a clear and useful discussion of the constant preservation issue, which is frequently overlooked until strange results show up. An additional useful criterion is the rate at which total tracer quantity is gained or lost. This will impact the ability to get a physically meaningful equilibrium solution.
  - ○ Looking back at this, we agree that this would be a useful metric. However, our goal when writing this section was to comment on how well constant preservation is maintained, even under single precision arithmetic and aggressive optimization. Since the manuscript was originally submitted, compute systems have changed and we have had to scrub simulation data to save on storage. To produce this metric at this point would require some time and effort that we currently do not have the funding for at the moment.

- @ 203: The use of single precision for the parent model will limit the usefulness of the results. It also makes it difficult to distinguish the sources of errors in the analysis.
  - ○ The parent model is run in double precision, but the transport operators are stored as single precision. It is not clear to us why single precision should limit the usefulness of results. The Constant Preservation results provide some measure of round-off error noise in the volume field (2-3 orders of magnitude smaller than surface values after 5 years of integration). Errors in an $O(1)$ constant tracer field are at worst 3-4 orders of magnitude smaller than the expected answer (a constant field). You are correct in that we cannot decipher between numerical and round-off error based on our results; this also cannot be done if we were just running with double precision. Perhaps the comparison of single to double precision is a topic for another study.

- @212 this paragraph: Clarify that the error data is from running the global or regional version of the offline model. There appears to be an assumption that these errors are not present in the parent dynamic model. It would be instructive to run this test on the dynamic model, in order to compare the error level inherent in the parent dynamic model.
    - The discretization for tracers in POP and in the offline volume preserve constant tracers. We've added an theoretical analysis in this Methodology section that shows a constant initial condition will remain a constant under the discretization when using exact arithmetic and in combination with the time integrators presented. The source of error for the constant preservation test case arises from floating point arithmetic, which we characterize in perhaps the most catastrophic scenario - single precision arithmetic with aggressive optimizations.
    - We agree that it would be useful to perform this conservation test with the parent model for comparison. Unfortunately, the machine that we used for these simulations has been decommissioned, and there are no plans to port E3SMv0-HiLAT to the new machine at LANL.

- @ 222: …"with and without mixing": vertical mixing? Please clarify.
    - The additional analysis described above now clarifies "with and without mixing". The intention of showing this in the constant preservation case is to highlight how variable coefficient vertical mixing can lead to an amplification of round-off errors above 1000m. We feel it's important to understand this behavior.

- @233 How does the 1000m division relate to the mixed-layer depth of the model? In Figures 4 and 5, Dye 4, especially, raises the question about the depth of the mixed layer in this vicinity.

    - The figure below shows the maximum mixed layer depth in the Argentine Basin in simulation year 64 during the month of September, which is the month when the mixed layer is deepest. Clearly the mixed layer is quite deep, but it does not reach as deep as 1000 m. We included the following statement in "The Argentine Basin Test Problem" section:

        *Note that maximum mixed layer depths in the Argentine Basin in winter are around 500 m in this model, so deep convection should not play a role in the transport of these tracers across the 1000 m depth horizon.*

[Figure]

- Figure 4: The scales on the figures are not all the same. This creates the visual impression that the small contributions are more important than they are, compared to the major contributors. Figure 5: Interesting set of graphs. Again, the scales on the graphs are very different, creating a visual that is out of proportion for the more slowly arriving contributors.

    - From an oceanographic perspective it is certainly relevant that there is an almost two orders of magnitude difference in the inventories of the 6 dye tracers, depending on their release location. But that is not the point that we are trying to make here. The purpose of Figs. 4 and 5 is to compare tracer inventories produced by the online and offline methods, and the color and axis scales were chosen to optimize the information content of these plots. Plotting these inventories on the same scale would significantly diminish -or even nullify- the value that these plots may have about the online/offline comparison for the less abundant dye tracers.

- @340 Supply a reference for the statement : …"reduces to a first order upwind scheme".
    - To address this comment, after changing the order of some of these sections, we've moved this discussion to a new section following the Graph Coloring to Operator Diagnosis section. We've added statements that describe how a TVD flux limiter works, with references, to illustrate this point.

- @361 Need to add finding the equilibrium solution under future work.
  - The conclusions paragraph has been expanded to incorporate this note.

- @368 Your conclusion that the analysis "highlights the limitations of the IRF approach", considering the limitations of this particular implementation, might be more appropriately stated as "highlights the challenges of the IRF approach".
  - This change has been made in the conclusions and abstract.

- @ 6  You don't say the nature of these benefits: is your method faster ? More accurate? Also, I don't know if "clearly shows" is the best phrasing, because it is not determined whether you approach is more accurate than the online approach (it seems more diffusive) and in terms of time, as far as I understood, the comparison is done on estimated costs and not actually running the online vs. offline head to head.
  - We understand your point. In other responses below, we address why online v. offline head-to-head comparison is not done and why we have chosen to rely on an estimate. The purpose of this paper is to show progress in FEOTS development as a toolkit for offline tracer simulation. This part of the abstract now reads

    *The demonstration illustrates progress in developing offline passive tracer simulation capabilities, while highlighting the challenges of the Impulse Response Functions approach in capturing tracer transports by a non-linear advection scheme. Our future work will focus on improving the computational efficiency of the code to reduce time-to-solution, using different basis functions to better represent non-linear advection operators, applying FEOTS to a parent model with unstructured grids (MPAS-Ocean), and to fully implement a Newton-Krylov steady state solver.*

- @ 35 I think you should spell out what the state of the art approach is and in what way your method improves it. Your narrative is a little too vague

  - We now discuss that existing TTM models have been applied to non-eddying ocean states only, and that FEOTS is specifically designed to tackle the large computational problems associated with tracer transport in a global eddying ocean.

- @ 45 Although above you said that climate models use parameterizations to describe the presence of eddies, can you clarify?

- - We agree that the logic of these sections did not work well, and we rewrote this section. We argue that low-resolution GCMs cannot represent processes like those observed in the Argentine Basin, and that the current generation of TTM models, which are based on low-resolution GCMs, are therefore inadequate tools to study tracer mixing in the Argentine Basin.

- @ 50 Could you expand on the current state of the art for the IRF approach and say more on what the differences with your work are ?

  - - We rewrote this section, also in response to the reviewer's previous comments, as we argue that the main innovations of FEOTS is its capabilities to diagnose operators using a method that requires fewer impulse fields (using graph coloring), to solve problems associated with eddying ocean states, and to provide capabilities for modeling regional subdomains of a parent model.

- @ 60 You may want to rephrase this and just say what the magnitude of the error is (like 10^-something)
  - - We've made the suggested change.

- @ 65 What application ?
  - - FEOTS ; the phrasing of these sentences felt awkward to end the previous sentence with FEOTS and start the next sentence immediately with "FEOTS" though we understand the confusion this has caused. This section has been modified to help improve clarity.

- @ 79 What discretization method are you using ? where are the unknowns located on the mesh ? what type of cells are you using ? The section on the Parent Model should be moved up so this sentence is put into context for the reader.
  - - The Parent model section has been moved to the first subsection of the methodology section; this was also requested by the other reviewer. We have reiterated at this location that, for our example discussed in this paper, we are using the Flux-Limited Lax-Wendroff advection scheme on POP's Arakawa B-Grid.

- @84 Why is your offline approach better than just discretizing the operators and coding them up ? What is the accuracy of this method ? Has it been tested against toy problems for which there was an analytical solution? If so, proper references should be cited here. If not, at least a heuristical discussion on accuracy should be added.
  - - Our approach captures the parent model discretizations along with the fluid velocity fields. Diagnosing operators with FEOTS assistance only requires knowledge of the advection and diffusion stencils used in the parent model. For linear discretizations (with respect to the tracer fields), the sparse matrix that is diagnosed is an exact representation of the advection and diffusion subroutines in the parent model. This methodology is demonstrated in the two references

cited in the highlighted sentence. When diagnosing operators from nonlinear flux-limiting/TVD advection schemes, the methodology will diagnose a more diffusive upwind advection operator than what the parent model advection scheme is capable of.  A discussion of this issue has been added to the following sections
- Parent Model
- A new section following the Graph Coloring approach to Operator Diagnosis section

The methodology we have presented is meant to be functional for a range of parent models, not just POP. Given a mesh (structured or unstructured) and the computational stencil that reflects the domain of influence of the underlying discretizations, FEOTS provides the infrastructure to diagnose an existing model's advection and diffusion schemes. The "glue" between FEOTS and a parent model is made up of a code layer that reads in a parent model's mesh and translates the connectivity information into a standard undirected graph representation. Once in this form, FEOTS can map between sparse matrix and 1-D vector representations and the parent models' memory layout. This allows us to use the same set of routines for offline simulation, independent of the parent model. In our view, this significant amount of code reuse reduces code maintenance as community models (that we would use as parent models) fall in and out of popularity.

Another benefit of capturing transport operators as sparse matrices is that we will be able to lean on linear algebra packages to assist us with future development. Given that we have proof-of-concept for our OO Fortran implementation of this methodology, we clearly have some work to do on improving time-to-solution. Packages like PSBLAS will help us expose parallelism in SpMV operations as well as implement solvers that will be necessary for tracer equilibration, which we plan to implement in future releases.

We have updated the introduction to incorporate these kinds of statements.

- @ 87 Please define in mathematical terms what you mean by "tendency of the impulse function".
  - The tendency is the time rate of change of the tracer field; it is equivalent to the impulse response function. We've rephrased this sentence and the equation below to provide a mathematical definition of the impulse response function.

- @ 89 Is Eq. (14) a better way to write this ? If A is the operator you are trying to estimate and R is known data, you should probably just write it as in Eq. (14).
  - Equation (14) is a good way to write this; this section has been rewritten, incorporating your other comments and those of the other reviewer to clarify the

connections between the impulse response function and the matrix representation of the transport operator.

- @ 100 Does this procedure depend on the time step? In the fully discretized system, the entries of A are multiplied by delta t (time-step) I believe. Do you need to estimate again A any time the delta t for your simulation changes ?
  - In short, this procedure does not depend on the time step. The changes made in response to your previous comments should clarify this.

- @ 108-109 Is this the main difference between your approach and Bardin 2014 ? Why 125 tracers and not another number ?
  - This is one difference. We also are focusing on eddy-permitting and higher resolution simulations which has also motivated developing a software package in a compiled language. In Bardin 2014, their parent model used an advection scheme that extends two grid-cells in each spatial dimension, giving a 5x5x5 brick (125 grid points). Bardin 2014, like others who have followed this methodology (e.g. Khatiwala), simply defined "checkerboard" fields for the impulse functions, using a formula like we show immediately below this comment.

    We've added the statement "... giving a 5x5x5 brick for the domain of influence." to this sentence to make it clearer why 125 tracers.

- @ 130 Who is v ? it was not in Eq. (4)
  - v is the volume anomaly. Equation (4) has been adjusted to include the volume anomaly correction. The text beneath equation set (4) now describes the volume anomaly term and why it is included (with a reference to the POP manual).

    We've also added a description for v in the Time integration section and why it is calculated the way we do. Additionally, we've added exposition in the Constant Preservation results, per the other reviewer's request that shows how this formulation conserves total tracer.

- @ 130 How are the volume anomalies discretized ? if one wanted to reproduce what you do, how do they deal with the volume anomalies ?
  - The volume anomaly is a discrete field that resides on the tracer grid points of the Arakawa B-grid. The source of volume anomalies is from the non-zero vertical velocities at the ocean free surface. The modeled tracer field, both in the "online" parent model and in the "offline" FEOTS simulation, is a tracer concentration ( a per unit-volume metric ). We want to be able to conserve the total amount of tracer ( volume multiplied by tracer concentration. Equations 9 and 10 show how the volume anomalies are advanced in time and incorporated into the tracer equation.

- @ 149 Can you clarify the differences between this work and Bardin 2014 ? Up to this point it sounds like you are following their approach almost entirely.

- ○ The main differences between are work and Bardin 2014 are as follows
    - ■ Our focus is on providing a toolkit for offline eddy-permitting/eddy-resolving simulations of passive tracers; Bardin (2014) focused on 1 deg resolution simulations. Global simulations with FEOTS are currently slow, but this paper aims to highlight some progress we have made in the initial version of FEOTS.
    - ■ FEOTS provides a suite of compiled Fortran programs to assist in impulse function creation, sparse matrix diagnosis from the impulse and impulse response functions, and offline tracer simulation. It is written using OO principles to allow extensibility to other parent models. This is in comparison to a monolithic MATLAB implementation.  We opted to go with a compiled language for performance reasons; removing JIT compiling provides performance gains; the ability to eventually parallelize this code through domain decomposition / parallel sparse matrix libraries will help us tackle global offline eddying simulations.
    - ■ Our graph coloring methodology generalizes the impulse field creation.
    - ■ FEOTS allows for regional simulations from a parent model. Methods are provided to extract the rows and columns of a transport operator for a subdomain of the parent model. This feature is demonstrated in the example simulations presented.
    - ■ The remainder of the offline modeling approach is identical to Bardin 2014
  Statements have been added in the introduction to this effect to show how our work is distinguished from Bardin 2014 and other predecessors.

- ● @ 149 I think it makes sense to move this section at the beginning of the paper
    - ○ We have moved this section further up in the paper; reviewer #1 made a similar request.

- ● @ 169 In your notation throughout the paper A is almost always a matrix, which is a linear operator, so (12) would always hold, maybe use a different notation like A(c1 + c2) \neq A(c1) + A(c2) ?
    - ○ We understand the confusion the notation here has caused. The main point we wanted to get across is that the parent model uses a nonlinear advection scheme and the offline model assumes the advection scheme can be written as a linear matrix vector multiplication; in this case, the two are not equivalent. In the presence of smooth tracer fields, the flux-limited Lax Wendroff method is third order accurate and has little diffusion (the leading truncation error is biharmonic). In fields that are not locally monotonic (like our impulse functions), the method reduces to an upwind method that introduces numerical diffusion. We now include discussion that illustrates how the impulse fields result in diagnosing advection operators that are more diffusive than the With requests from the other reviewer to clarify how the flux-limiting results in a transport operator that is more diffusive than Lax-Wendroff (without flux-limiting), we've rewritten these statements about the issues with using nonlinear advection schemes with this

approach. Because the parent model description now comes before the description of the offline model and the operator diagnosis, we've moved this discussion to a new section following the "Graph Coloring to Operator Diagnosis" section.

- @ 200 The FV discretization should be mentioned when you bring up the semi-discrete model
  - We've moved the parent model description before this section and have made changes to define what methods are used to discretize the advection operator. This should clear up this issue.

- @ 203 But is it a physically reasonable choice ? Also, the way you test accuracy is potentially hindered by having a smaller number of significant digits.
  - This is a question that is open for debate and it is not a question this paper aims to address. The focus of this paper is to present to the community a tool for offline tracer simulation and share progress and current capabilities. This example of constant tracer preservation is unique in that numerical errors are zero (proof of this is now included in the paper). Any deviation from a constant value for the tracer is a result of round-off errors accumulating over time. We present a "worst-case scenario" for round-off errors (single precision and aggressive optimization). What we show in these figures is that the error in an $O(1)$ constant tracer field is at worst 3-4 orders of magnitude smaller than the expected answer (a constant field equal to 1). As we indicated to the other reviewer, perhaps the comparison of single to double precision is a topic for another study. Defining physically reasonable comes down to what fields we are trying to measure and with what confidence.

- @ 216-218 Do you consider these to be small errors ? for single precision, machine precision is around $10^{-8}$ and here it looks like you are several orders of magnitude above that. Can you comment ?
  - The -Ofast flag in gfortran enables a number of unsafe math operations. From the GNU documentation :

    -Ofast
    Disregard strict standards compliance. -Ofast enables all -O3 optimizations. It also enables optimizations that are not valid for all standard-compliant programs. It turns on -ffast-math, -fallow-store-data-races and the Fortran-specific -fstack-arrays, unless -fmax-stack-var-size is specified, and -fno-protect-parens. It turns off -fsemantic-interposition.

    In our experience, Ofast incurs larger round-off errors than machine epsilon for single precision numbers - why this happens may best be answered by a compiler writer.

When using the five day averaged operators, as we are in this case, the transport operators are held fixed over 5 days of simulation time. The volume anomaly grows linearly with time when the operator is held fixed; this includes non-surface layers where the only contributing non-zero values are given by round-off error. Our working idea here is that the round-off errors accumulate over time.

- @ 228 Please add an explanation to describe exactly what you mean by "online"

  We now add the clarification:

  *Do the tracer distributions simulated by FEOTS using transport operators diagnosed from the parent model E3SMv0-HiLAT03 (offline) faithfully represent the tracer distributions simulated by the parent model itself (online)?*

- @ 236 What do you mean with "inventory"

  We replaced the term *inventory* by *stock* to better express that we are discussing the total amount of dye tracer integrated over a region.

- @ 288 Can you comment on how much the nature of the problem affects the averaging methodology ?
  - We have added the following to the beginning of this section

    *The parent model is capable of producing velocity fields that have a wide range of scales of spatial and temporal variability. The shortest temporal periods are on the order of a few time steps and the longest period is the duration of the simulation. In general, higher resolution models introduce more variability on shorter length and time scales and some consideration is needed when selecting an averaging period for the transport operator diagnosis. For storage reasons, it is not practical to store snapshots of the transport operators at every time step. Conversely, representing the ocean transport with long time averages may exclude the effects of important variability. The choice in time averaging period for the transport operators can impact the evolution of tracers calculated in FEOTS and an appropriate balance of practicality and accuracy should be struck.*

- @ 290 Then maybe this paragraph should be the first one you discuss in your results section
  - We've rephrased the introduction to this section to read as

    *One goal of FEOTS is to perform tracer calculations at a lower computational cost than the parent model. Additionally, FEOTS allows researchers to take advantage of transport operators produced by state-of-the-art climate simulations to conduct regional offline simulations. This provides flexibility in studying ocean*

*transport phenomena and increases the value of online produced model data while considerably reducing the computational expense for researchers solely interested in studies involving passive tracers. Here we evaluate the computational performance of a regional FEOTS configuration and compare it with the global parent model.*

- @ 298 - 299 Can you clarify how the averaging processes works ? For instance, considering your 1 day and 5 day averaged operators, how many runs did you do for each of them ?
  - We've added more details to the Graph Coloring approach to operator diagnosis that illustrates how the impulse response functions are diagnosed and time averaged.

- Table 3 Would it be possible to actually do a run where you report head to head the times for online and offline in seconds to clearly show which one is faster ? Estimated costs do not provide strong enough grounds in my opinion to claim that the benefits of your approach are clear.
  - We have experimented some with running FEOTS in a global configuration and have found that the simulation is fairly slow, although we don't have direct measurements recorded. Additionally, the memory requirement is about 15 GB in single precision ( 30 GB in double precision ). For memory bound algorithms (like finite difference, finite volume, and sparse matrix vector multiplication) this is quite a bit of load to put on a single core. We've rephrased some things in this section to highlight these facts as well as to indicate that we are suggesting that it is probable that a global configuration of FEOTS could provide comparable runtimes to the parent model with fewer CPU-hours. This, to us, suggests viability and motivates us to explore exposing parallelism in FEOTS' core methods.

    We have updated the text in this section to reflect these remarks.

- @ 331 This is why I believe a head to head comparison would be more informative.
  - We had started a few global simulations during this study, but found that the time-to-solution was too high; we simply would not have had enough time to complete this study in the time-frame allotted for this work. This is because FEOTS is in serial; the hotspot profile shows where most of the time is spent while running the code in forward-integration mode. However, our goal in this section was to show the amount of compute resources ( number of cores multiplied by the time spent using them ) is lower for FEOTS; if this weren't the case, parallelizing would provide no hope of getting the time to solution lower than the parent model and would invalidate our whole motivation for doing this. This particular statement you've highlighted was mentioned to indicate where our focus is going to be in the near future in terms of improving time-to-solution.

- @ 356 Why did you do an estimate of the time instead of actually running offline vs online head to head ?
  - FEOTS is currently written in serial and is considerably slower, at the moment, than running POP with O(1000) MPI Ranks. We simply did not have the time to run such a long serial simulation to go "head to head". However, our estimations suggest that the amount of compute resources required to run FEOTS globally would be lower than the online parent model. This is encouraging when considering whether or not to put effort into parallelizing FEOTS.